# Agentic Context Engineering: Evolving Contexts for Self-Improving Language Models

**Qizheng Zhang**[1*], **Changran Hu**[2*], **Shubhangi Upasani**[2], **Boyuan Ma**[2], **Fenglu Hong**[2],
**Vamsidhar Kamanuru**[2], **Jay Rainton**[2], **Chen Wu**[2], **Mengmeng Ji**[2], **Hanchen Li**[3],
**Urmish Thakker**[2], **James Zou**[1], **Kunle Olukotun**[1]

[1]Stanford University    [2]SambaNova Systems, Inc.    [3]UC Berkeley

{qizhengz,kunle}@stanford.edu    changran_hu@berkeley.edu

 ace-agent/ace     ace-agent.github.io    *Equal contribution.

## Abstract

Large language model (LLM) applications such as agents and domain-specific reasoning increasingly rely on *context adaptation*: modifying inputs with instructions, strategies, or evidence, rather than weight updates. Prior approaches improve usability but often suffer from brevity bias, which drops domain insights for concise summaries, and from context collapse, where iterative rewriting erodes details over time. We introduce ACE (**A**gentic **C**ontext **E**ngineering), a framework that treats contexts as evolving playbooks that accumulate, refine, and organize strategies through a modular process of generation, reflection, and curation. ACE prevents collapse with structured, incremental updates that preserve detailed knowledge and scale with long-context models. Across agent and domain-specific benchmarks, ACE optimizes contexts both offline (*e.g.,* system prompts) and online (*e.g.,* agent memory), consistently outperforming strong baselines: +10.6% on agents and +8.6% on finance, while significantly reducing adaptation latency and rollout cost. Notably, ACE could adapt effectively without labeled supervision and instead by leveraging natural execution feedback. On the AppWorld leaderboard, ACE matches the top-ranked production-level agent on the overall average and surpasses it on the harder test-challenge split, despite using a smaller open-source model. These results show that comprehensive, evolving contexts enable scalable, efficient, and self-improving LLM systems with low overhead.

## 1 Introduction

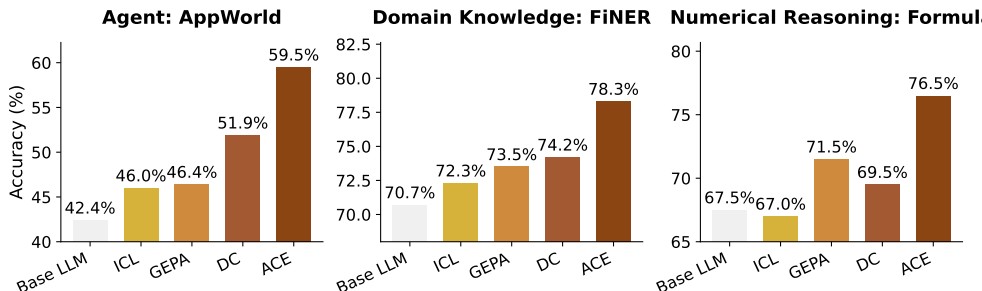

Figure 1: **Overall Performance Results.** Our proposed framework, ACE, consistently outperforms strong baselines across agent and domain-specific tasks.

Modern AI applications based on large language models (LLMs), such as LLM agents (Yao et al., 2023; Yang et al., 2024) and compound AI systems (Zaharia et al., 2024), increasingly depend on *context adaptation*. Instead of modifying model weights, context adaptation improves performance after model training by incorporating clarified instructions, structured reasoning steps, or domain-

specific input formats directly into the model's inputs. Contexts underpin many AI system components, including system prompts that guide downstream tasks (Opsahl-Ong et al., 2024; Agrawal et al., 2025), memory that carries past facts and experiences (Suzgun et al., 2025; Xu et al., 2025), and factual evidence that reduces hallucination and supplements knowledge (Asai et al., 2024).

Adapting through *contexts* rather than *weights* offers several key advantages. Contexts are interpretable and explainable for users and developers (Wei et al., 2022; Wang et al., 2023), allow rapid integration of new knowledge at runtime (Lewis et al., 2020; Borgeaud et al., 2022), and can be shared across models or modules in a compound system (Khot et al., 2023). Meanwhile, advances in long-context LLMs (Peng et al., 2024) and context-efficient inference such as KV cache reuse (Gim et al., 2024; Yao et al., 2025) are making context-based approaches increasingly practical for deployment. As a result, context adaptation is emerging as a central paradigm for building capable, scalable, and self-improving AI systems.

Despite this progress, existing approaches to context adaptation face two limitations. First, *brevity bias*: many prompt optimizers prioritize concise applicable instructions over comprehensive accumulation. For example, GEPA (Agrawal et al., 2025) highlights brevity as a strength, but such abstraction can omit domain-specific heuristics, tool-use guidelines, or common failure modes that matter in practice (Gao et al., 2025). This objective aligns with validation metrics in some settings, but often fails to capture the detailed strategies required by agents and knowledge-intensive applications. Second, *context collapse*: methods that rely on monolithic rewriting by an LLM often degrade into shorter, less informative summaries over time, causing sharp performance declines (Figure 2). In domains such as interactive agents (Trivedi et al., 2024; Patil et al., 2024; Zhang et al., 2024), domain-specific programming (Ye et al., 2023; Zhang et al., 2025a;b; Mang et al., 2025), and financial or legal analysis (Loukas et al., 2022; Guha et al., 2023; Wang et al., 2025a), strong performance depends on retaining detailed, task-specific knowledge rather than compressing it away.

As applications like agents and knowledge-intensive reasoning demand greater reliability, recent work has shifted toward saturating contexts with abundant, potentially useful information (Jiang et al., 2025; Chung et al., 2025; Chen et al., 2025), enabled by advances in long-context LLMs (Peng et al., 2024; Mao et al., 2024). **We argue that contexts should function not as concise summaries, but as comprehensive, structured playbooks that are detailed, inclusive, and rich with domain insights.** Unlike humans, who often benefit from concise generalization, LLMs are more effective when provided with long, detailed contexts and can distill relevance autonomously (Jiang et al., 2025; Liu et al., 2025b; Suzgun et al., 2025). Thus, instead of compressing away domain-specific heuristics and tactics, contexts should preserve them, allowing the model to decide what matters during inference time.

To address these limitations, we introduce ACE (**A**gentic **C**ontext **E**ngineering), a framework for comprehensive context adaptation in both offline settings (*e.g.,* system prompt optimization) and online settings (*e.g.,* test-time memory adaptation). Rather than compressing contexts into distilled summaries, ACE treats them as evolving playbooks that accumulate and organize strategies over time. By design, ACE incorporates a modular workflow of generation, reflection, and curation, while adding structured, incremental updates guided by a grow-and-refine principle. This design preserves detailed, domain-specific knowledge, prevents context collapse, and yields contexts that remain comprehensive and scalable throughout adaptation.

We evaluate ACE on two categories of LLM applications that most benefit from comprehensive, evolving contexts: (1) *agents* (Trivedi et al., 2024), which require multi-turn reasoning, tool use, and environment interaction, where accumulated strategies can be reused across episodes; and (2) *domain-specific benchmarks*, which demand specialized tactics and knowledge, like financial analysis (Loukas et al., 2022; Wang et al., 2025a). Our key findings are:

- ACE consistently outperforms strong baselines, yielding average gains of 10.6% on *agents* and 8.6% on *domain-specific benchmarks*, across both offline and online adaptation settings.
- ACE is able to construct effective contexts *without* labeled supervision, instead leveraging execution feedback and environment signals, key ingredients for self-improving LLMs and agents.
- On the AppWorld benchmark leaderboard (AppWorld), ACE surpasses the top-1-ranked production-level agent IBM-CUGA (Marreed et al., 2025) (powered by GPT-4.1) while using an open-source model (DeepSeek-V3.1).

- ACE requires significantly fewer rollouts and achieves lower adaptation latency than existing adaptive methods, demonstrating that scalable self-improvement can be achieved with both higher accuracy and lower cost.

## 2  BACKGROUND AND MOTIVATION

### 2.1  CONTEXT ADAPTATION

Context adaptation (or context engineering) refers to methods that improve model behavior by constructing or modifying inputs to an LLM, rather than altering its weights. The current state of the art leverages *natural language feedback* (Shinn et al., 2023; Yuksekgonul et al., 2025; Agrawal et al., 2025). In this paradigm, a language model inspects the current context along with signals such as execution traces, reasoning steps, or validation results, and generates natural language feedback on how the context should be revised. This feedback is then incorporated into the context, enabling iterative adaptation. Representative methods include Reflexion (Shinn et al., 2023), which reflects on failures to improve agent planning; TextGrad (Yuksekgonul et al., 2025), which optimizes prompts via gradient-like textual feedback; GEPA (Agrawal et al., 2025), which refines prompts iteratively based on execution traces and achieves strong performance, even surpassing reinforcement learning approaches in some settings; and Dynamic Cheatsheet (Krause et al., 2019), which constructs an external memory that accumulates strategies and lessons from past successes and failures during inference. These natural language feedback methods represent a major advance, offering flexible and interpretable signals for improving LLM systems beyond weight updates.

### 2.2  LIMITATIONS OF EXISTING CONTEXT ADAPTATION METHODS

**Brevity Bias**    A recurring limitation of context adaptation methods is *brevity bias*: the tendency of optimization to collapse toward short, generic prompts. Gao et al. (Gao et al., 2025) document this effect in prompt optimization for test generation, where iterative methods repeatedly produced near-identical instructions (*e.g.,*, "Create unit tests to ensure methods behave as expected"), sacrificing diversity and omitting domain-specific detail. This convergence not only narrows the search space but also propagates recurring errors across iterations, since optimized prompts often inherit the same faults as their seeds. More broadly, such bias undermines performance in domains that demand detailed, context-rich guidance—such as multi-step agents, program synthesis, or knowledge-intensive reasoning—where success hinges on accumulating rather than compressing task-specific insights.

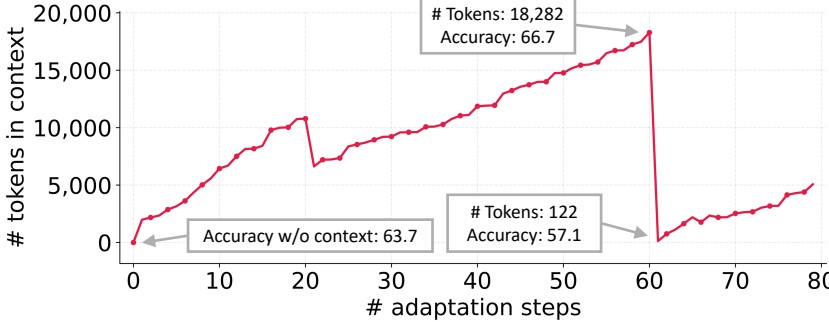

Figure 2: **Context Collapse.** Monolithic rewriting of context by an LLM can collapse it into shorter, less informative summaries, leading to sharp performance drops.

**Context Collapse**    In a case study on the AppWorld benchmark (Trivedi et al., 2024), we observe a phenomenon we call *context collapse*, which arises when an LLM is tasked with fully rewriting the accumulated context at each adaptation step. As the context grows large, the model tends to compress it into much shorter, less informative summaries, causing a dramatic loss of information. For instance, at step 60 the context contained 18,282 tokens and achieved an accuracy of 66.7, but at the very next step it collapsed to just 122 tokens, with accuracy dropping to 57.1—worse than the baseline accuracy of 63.7 without adaptation. While we highlight this through Dynamic Cheat-

sheet (Suzgun et al., 2025), the issue is not specific to that method; rather, it reflects a fundamental risk of end-to-end context rewriting with LLMs, where accumulated knowledge can be abruptly erased instead of preserved.

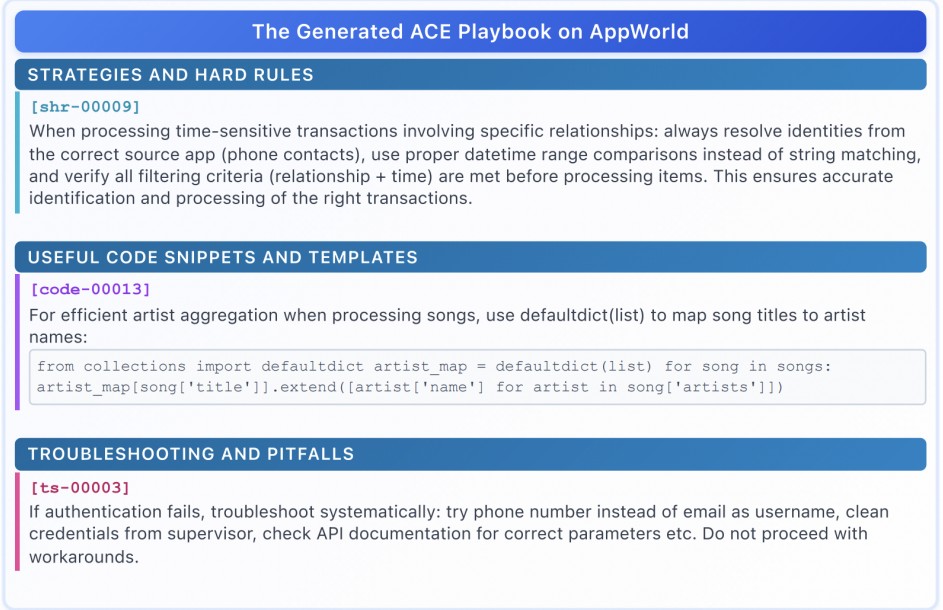

Figure 3: **Example ACE-Generated Context on the AppWorld Benchmark** (partially shown). ACE-generated contexts contain detailed, domain-specific insights along with tools and code that are readily usable, serving as a comprehensive playbook for LLM applications.

## 3 AGENTIC CONTEXT ENGINEERING (ACE)

We present ACE (**A**gentic **C**ontext **E**ngineering), a framework for scalable and efficient context adaptation in both offline (*e.g.,* system prompt optimization) and online (*e.g.,* test-time memory adaptation) scenarios. Instead of condensing knowledge into terse summaries or static instructions, ACE treats contexts as evolving playbooks that continuously accumulate, refine, and organize strategies over time. Inspired by the agentic design of Dynamic Cheatsheet (Suzgun et al., 2025), ACE introduces a structured division of labor across three roles (Figure 4): the *Generator*, which produces reasoning trajectories; the *Reflector*, which distills concrete insights from successes and errors; and the *Curator*, which integrates these insights into structured context updates. This mirrors how humans learn: experimenting, reflecting, and consolidating, while avoiding the bottleneck of overloading a single model with all responsibilities.

To address the limitations of prior methods discussed in §2.2 (notably *brevity bias* and *context collapse*) ACE introduces three key innovations: (1) a dedicated *Reflector* that separates evaluation and insight extraction from curation, improving context quality and downstream performance (§4.6); (2) incremental *delta updates* (§3.1) that replace costly monolithic rewrites with localized edits, reducing both latency and compute cost (§4.7); and (3) a *grow-and-refine* mechanism (§3.2) that balances steady context expansion with redundancy control.

As shown in Figure 4, the workflow begins with the Generator producing reasoning trajectories for new queries, which surface both effective strategies and recurring pitfalls. The Reflector critiques these traces to extract lessons, optionally refining them across multiple iterations. The Curator then synthesizes these lessons into compact *delta entries*, which are merged deterministically into the existing context by lightweight, non-LLM logic. Because updates are itemized and localized, multiple deltas can be merged in parallel, enabling batched adaptation at scale. ACE further supports multi-epoch adaptation, where the same queries are revisited to progressively strengthen the context.

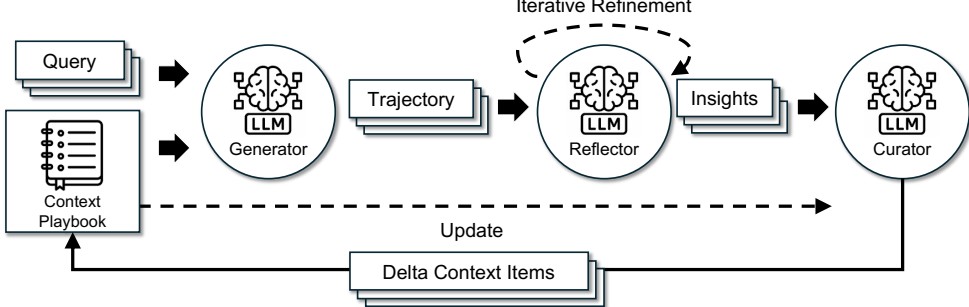

Figure 4: **The ACE Framework.** Inspired by Dynamic Cheatsheet, ACE adopts an agentic architecture with three specialized components: a Generator, a Reflector, and a Curator.

## 3.1 INCREMENTAL DELTA UPDATES

A core design principle of ACE is to represent context as a collection of *structured, itemized bullets*, rather than a single monolithic prompt. The concept of a bullet is similar to the concept of a memory entry in LLM memory frameworks like Dynamic Cheatsheet (Suzgun et al., 2025) and A-MEM (Xu et al., 2025), but builds on top of that and consists of (1) **metadata**, including a unique identifier and counters tracking how often it was marked helpful or harmful; and (2) **content**, capturing a small unit such as a reusable strategy, domain concept, or common failure mode. When solving new problems, the Generator highlights which bullets were useful or misleading, providing feedback that guides the Reflector in proposing corrective updates.

This itemized design enables three properties: (1) *localization*, so only the relevant bullets are updated; (2) *fine-grained retrieval*, so the Generator can focus on the most pertinent knowledge; and (3) *incremental adaptation*, allowing efficient merging, pruning, and de-duplication during inference.

Rather than regenerating contexts in full, ACE incrementally produces compact *delta contexts*: small sets of candidate bullets distilled by the Reflector and integrated by the Curator. This avoids the computational cost and latency of full rewrites, while ensuring that past knowledge is preserved and new insights are steadily appended. As contexts grow, this approach provides the scalability needed for long-horizon or domain-intensive applications.

## 3.2 GROW-AND-REFINE

Beyond incremental growth, ACE ensures that contexts remain compact and relevant through periodic or lazy refinement. In grow-and-refine, bullets with new identifiers are appended, while existing bullets are updated in place (*e.g.,* incrementing counters). A de-duplication step then prunes redundancy by comparing bullets via semantic embeddings. This refinement can be performed proactively (after each delta) or lazily (only when the context window is exceeded), depending on application requirements for latency and accuracy.

Together, incremental updates and grow-and-refine maintain contexts that expand adaptively, remain interpretable, and avoid the potential variance introduced by monolithic context rewriting.

## 4 RESULTS

Our evaluation of ACE shows that:

- **Enabling High-Performance, Self-Improving Agents.** ACE enables agents to self-improve by dynamically refining their input context, both in offline and online settings. It boosts accuracy on the AppWorld benchmark by up to 17.1% by learning to engineer better contexts from execution feedback alone, without needing ground-truth labels. (§4.3)
- **Large Gains on Domain-Specific Benchmarks.** On complex financial reasoning benchmarks, ACE delivers an average performance gain of 8.6% over strong baselines by constructing comprehensive playbooks with domain-specific concepts and insights. (§4.4)

- **Effective by Design.** Ablation studies confirm our design choices are key to success, with components like the Reflector, multi-epoch refinement, and incremental delta update each contributing substantial performance gains. (§4.6)
- **Lower Cost and Adaptation Latency.** ACE achieves these gains efficiently, reducing adaptation latency by 86.9% on average, while requiring fewer rollouts and lower token dollar costs. (§4.7)

## 4.1 TASKS AND DATASETS

We evaluate ACE on two categories of LLM applications that benefit most from evolving contexts: (1) *LLM agent*, which require multi-turn reasoning, tool use, and environment interaction; with ACE, agents can accumulate and reuse strategies across episodes and environments; and (2) *domain-specific reasoning*, which demand mastery of specialized concepts and tactics; we focus on financial analysis as a main case study, and show additional results on medical reasoning and text-to-SQL.

- **LLM Agent: AppWorld (Trivedi et al., 2024)** is a suite of autonomous agent tasks involving API understanding, code generation, and environment interaction. It provides a realistic execution environment with common applications and APIs (*e.g.,* email, file system) and tasks of two difficulty levels (normal and challenge). A public leaderboard (AppWorld) tracks performance, where, at the time of submission, the best system achieved only 60.3% average accuracy, highlighting the benchmark's difficulty and realism.
- **Domain-Specific Reasoning: Financial, Medical, and Text-to-SQL Benchmarks** We use finance as our main case study in §4.4. For financial analysis, we focus on FiNER (Loukas et al., 2022) and Formula (Wang et al., 2025a), which test LLMs on financial reasoning tasks that rely on the eXtensible Business Reporting Language (XBRL). *FiNER* requires labeling tokens in XBRL financial documents with one of 139 fine-grained entity types, a key step for financial information extraction in regulated domains. *Formula* focuses on applying financial concepts and performing computations to answer queries, *i.e.,* numerical reasoning. Beyond finance, we evaluate on two additional domain tasks from StreamBench (Wu et al., 2024): DDXPlus (Fansi Tchango et al., 2022) (medical reasoning) and BIRD-SQL (Li et al., 2023) (text-to-SQL).

**Evaluation Metrics** For AppWorld, we follow the official benchmark protocol and report *Task Goal Completion* (TGC) and *Scenario Goal Completion* (SGC) on both the test-normal and test-challenge splits. For FiNER, Formula and DDXPlus, we follow the original setup and report accuracy, measured as the proportion of predicted answers that exactly match the ground truth. For BIRD-SQL, we use GPT-4o-mini (OpenAI, 2024) under LLM-as-a-judge (Zheng et al., 2023).

All datasets follow the original train/validation/test splits. For *offline* context adaptation, methods are optimized on the training split and evaluated on the test split with pass@1 accuracy. For *online* context adaptation, methods are evaluated sequentially on the test split: for each sample, the model first predicts with the current context, then updates its context based on that sample. The same shuffled test split is used across all methods.

## 4.2 BASELINES AND METHODS

**Base LLM** The base model is evaluated directly on each benchmark without any context engineering, using the default prompts provided by dataset authors. For AppWorld, we follow the official `ReAct` (Yao et al., 2023) implementation released by the benchmark authors, and build all other baselines and methods on top of this framework.

**In-Context Learning (ICL) (Agarwal et al., 2024)** ICL provides the model with task demonstrations in the input prompt (few-shot or many-shot). This allows the model to infer the task format and desired output without weight updates. We supply all training samples when they fit within the model's context window; otherwise, we fill the window with as many demonstrations as possible.

**MIPROv2 (Opsahl-Ong et al., 2024)** MIPROv2 is a popular prompt optimizer for LLM applications that works by jointly optimizing system instructions and in-context demonstrations via bayesian optimization. We use the official DSPy implementation (DSPy, b), setting `auto="heavy"` to maximize optimization performance.

**GEPA (Agrawal et al., 2025)** GEPA (Genetic-Pareto) is a sample-efficient prompt optimizer based on reflective prompt evolution. It collects execution traces (reasoning, tool calls, intermediate outputs) and applies natural-language reflection to diagnose errors, assign credit, and propose prompt updates. A genetic Pareto search maintains a frontier of high-performing prompts, mitigating local optima. Empirically, GEPA outperforms reinforcement learning methods such as GRPO and prompt optimizers like MIPROv2, achieving up to 10–20% higher accuracy with as much as 35× fewer rollouts. We use the official DSPy implementation (DSPy, a), setting `auto="heavy"` to maximize optimization performance.

**Dynamic Cheatsheet (DC) (Suzgun et al., 2025)** DC is a test-time learning approach that introduces an adaptive external memory of reusable strategies and code snippets. By continuously updating this memory with newly encountered inputs and outputs, DC enables models to accumulate knowledge and reuse it across tasks, often leading to substantial improvements over static prompting methods. A key advantage of DC is that it does not require ground-truth labels: the model can curate its own memory from its generations, making the method highly flexible and broadly applicable. We use the official implementation released by the authors (Suzgun et al.) and set it to use the `cumulative` mode (DC-CU).

**ACE (ours)** ACE optimizes LLM contexts for both offline and online adaptation through an agentic context engineering framework. To ensure fairness, we use the same LLM for the Generator, Reflector, and Curator (non-thinking mode of DeepSeek-V3.1 (Liu et al., 2024a)), preventing knowledge transfer from a stronger Reflector or Curator to a weaker Generator. This isolates the benefit of context construction itself. We additionally evaluate ACE with other backbone LLMs in the appendix, where we observe consistent gains. We adopt a batch size of 1 (constructing a delta context from each sample). We set the maximum number of Reflector refinement rounds and the maximum number of epoch in offline adaptation to 5.

## 4.3 RESULTS ON AGENT BENCHMARK

| Method | GT Labels | Test-Normal | | Test-Challenge | | Average |
|---|---|---|---|---|---|---|
| | | TGC↑ | SGC↑ | TGC↑ | SGC↑ | |
| DeepSeek-V3.1-671B as Base LLM | | | | | | |
| ReAct | | 63.7 | 42.9 | 41.5 | 21.6 | 42.4 |
| Offline Adaptation | | | | | | |
| ReAct + ICL | ✓ | $64.3_{+0.6}$ | $46.4_{+3.5}$ | $46.0_{+4.5}$ | $27.3_{+5.7}$ | $46.0_{+3.6}$ |
| ReAct + GEPA | ✓ | $64.9_{+1.2}$ | $44.6_{+1.7}$ | $46.0_{+4.5}$ | $30.2_{+8.6}$ | $46.4_{+4.0}$ |
| ReAct + ACE | ✓ | $\mathbf{76.2}_{+12.5}$ | $\mathbf{64.3}_{+21.4}$ | $\mathbf{57.3}_{+15.8}$ | $\mathbf{39.6}_{+18.0}$ | $\mathbf{59.4}_{+17.0}$ |
| ReAct + ACE | ✗ | $75.0_{+11.3}$ | $\mathbf{64.3}_{+21.4}$ | $54.4_{+12.9}$ | $35.2_{+13.6}$ | $57.2_{+14.8}$ |
| Online Adaptation | | | | | | |
| ReAct + DC (CU) | ✗ | $65.5_{+1.8}$ | $\mathbf{58.9}_{+16.0}$ | $52.3_{+10.8}$ | $30.8_{+9.2}$ | $51.9_{+9.5}$ |
| ReAct + ACE | ✗ | $\mathbf{69.6}_{+5.9}$ | $53.6_{+10.7}$ | $\mathbf{66.0}_{+24.5}$ | $\mathbf{48.9}_{+27.3}$ | $\mathbf{59.5}_{+17.1}$ |

Table 1: **Results on the AppWorld Agent Benchmark (DeepSeek-V3.1-671B as the Base LLM).** "GT labels" indicates whether ground-truth labels are available to the Reflector during adaptation. We evaluate the ACE framework against multiple baselines on top of the official `ReAct` implementation, both for offline and online context adaptation. `ReAct + ACE` outperforms selected baselines by an average of 10.6%, and could achieve good performance even without access to GT labels.

**Analysis: AppWorld** As shown in Table 1, ACE consistently improves over strong baselines on AppWorld. In the offline setting, `ReAct + ACE` outperforms both `ReAct + ICL` and `ReAct + GEPA` by significant margins (12.3% and 11.9%, respectively), demonstrating that structured, evolving, and detailed contexts enable more effective agent learning than fixed demonstrations or single optimized instruction prompts. These gains extend to the online setting, where ACE continues to outperform prior adaptive methods such as Dynamic Cheatsheet by an average of 7.6%.

In the agent use case, ACE remains effective even *without* access to ground-truth labels during adaptation: `ReAct + ACE` achieves an average improvement of 14.8% over the `ReAct` baseline

in this setting. This robustness arises because ACE leverages signals naturally available during execution (*e.g.,* code execution success or failure) to guide the Reflector and Curator in forming structured lessons of successes and failures. Together, these results establish ACE as a strong and versatile framework for building self-improving agents that adapt reliably both with and without labeled supervision.

Notably, on the latest AppWorld leaderboard (as of September 20, 2025; Figure 5), `ReAct` + ACE (59.4% average) matches the top-1-ranked IBM CUGA (60.3%)[1], a production-level GPT-4.1–based agent (Marreed et al., 2025), despite using the much smaller open-source model DeepSeek-V3.1. With online adaptation, `ReAct` + ACE even surpasses IBM CUGA by 8.4% in TGC and 0.7% in SGC on test-challenge, underscoring the effectiveness of ACE in building comprehensive and self-evolving contexts for agents.

## 4.4 RESULTS ON DOMAIN-SPECIFIC BENCHMARK

| Method | GT Labels | FiNER (Acc↑) | Formula (Acc↑) | Average |
|---|---|---|---|---|
| `DeepSeek-V3.1 as Base LLM` | | | | |
| `Base LLM` | | 70.7 | 67.5 | 69.1 |
| `Offline Adaptation` | | | | |
| ICL | ✓ | $72.3_{+1.6}$ | $67.0_{-0.5}$ | $69.6_{+0.5}$ |
| MIPROv2 | ✓ | $72.4_{+1.7}$ | $69.5_{+2.0}$ | $70.9_{+1.8}$ |
| GEPA | ✓ | $73.5_{+2.8}$ | $71.5_{+4.0}$ | $72.5_{+3.4}$ |
| ACE | ✓ | $\mathbf{78.3_{+7.6}}$ | $\mathbf{85.5_{+18.0}}$ | $\mathbf{81.9_{+12.8}}$ |
| ACE | ✗ | $71.1_{+0.4}$ | $83.0_{+15.5}$ | $77.1_{+8.0}$ |
| `Online Adaptation` | | | | |
| DC (CU) | ✓ | $74.2_{+3.5}$ | $69.5_{+2.0}$ | $71.8_{+2.7}$ |
| DC (CU) | ✗ | $68.3_{-2.4}$ | $62.5_{-5.0}$ | $65.4_{-3.7}$ |
| ACE | ✓ | $\mathbf{76.7_{+6.0}}$ | $76.5_{+9.0}$ | $\mathbf{76.6_{+7.5}}$ |
| ACE | ✗ | $67.3_{-3.4}$ | $78.5_{+11.0}$ | $72.9_{+3.8}$ |

Table 2: **Results on Financial Analysis Benchmark (DeepSeek-V3.1-671B as the Base LLM).** "GT labels" indicates whether ground-truth labels are available to the Reflector during adaptation. With GT labels, ACE achieves consistent improvements in both offline and online settings, highlighting the advantage of structured and evolving contexts for domain-specific reasoning. However, we also observe that in the absence of reliable feedback signals (*e.g.,* ground-truth labels or execution outcomes), both ACE and other adaptive methods such as Dynamic Cheatsheet may degrade, suggesting that context adaptation depends critically on feedback quality.

**Analysis: Finance Benchmark** As shown in Table 2, ACE delivers strong improvements on financial analysis benchmarks. In the offline setting, when provided with ground-truth answers from the training split, ACE surpasses ICL, MIPROv2, and GEPA by clear margins (an average of 10.9%), showing that structured and evolving contexts are particularly effective when tasks require precise domain knowledge (*e.g.,* financial concepts, XBRL rules) that goes beyond fixed demonstrations or monolithic optimized prompts. In the online setting, ACE continues to exceed prior adaptive methods such as DC by an average of 6.2%, further confirming the benefit of agentic context engineering for accumulating reusable insights across specialized domains.

Moreover, we also observe that when ground-truth supervision or reliable execution signals are absent, both ACE and DC may degrade in performance. In such cases, the constructed context can be polluted by spurious or misleading signals, highlighting a potential limitation of inference-time adaptation without reliable feedback. This suggests that while ACE is robust under rich feedback (*e.g.,* code execution results or formula correctness in agent tasks), its effectiveness depends on the

---

[1]We mention IBM CUGA as a rough contextual reference to show that ACE operates in a similar performance range on the AppWorld leaderboard. It is not used as a methodological baseline, and we do not make direct comparisons. CUGA's internal design differs from ACE's context-adaptation focus, and all baselines are evaluated under identical setups to isolate methodological effects rather than agent-engineering choices.

availability of signals that allow the Reflector and Curator to make sound judgments. We return to this limitation in §5.

**Analysis: Medical and Text-to-SQL Benchmark**  While this subsection focuses on finance as a detailed case study, ACE is not finance-specific: we also see consistent gains on other domain-specific tasks, including medical reasoning and text-to-SQL, suggesting that the same playbook-style context adaptation transfers across domains. Full results are reported in Appendix §A.2.

## 4.5 GENERALIZATION ACROSS LLMS

Table 1 and Table 2 use our default backbone (DeepSeek-V3.1), but ACE is not specific to this model. We can swap in other LLMs without changing the algorithm or prompts, and still see consistent gains on AppWorld and Finance benchmarks. Appendix §A.1 reports full results on GPT-OSS-120B, GPT-5.1, and Llama-3.3-70B-Instruct, where ACE improves over the corresponding base agents or models. These results suggest ACE is a generalizable method for test-time context evolution across LLM families.

## 4.6 ABLATION STUDY AND SENSITIVITY ANALYSIS

**Ablation Study**  Table 3 reports ablation studies on AppWorld, analyzing how individual design choices of ACE contribute to effective context adaptation. We examine three factors: (1) *the Reflector with iterative refinement*, our addition to the agentic framework beyond Dynamic Cheatsheet, (2) *multi-epoch adaptation*, which refines contexts over training samples multiple times, and (3) *offline warmup*, which initializes the context through offline adaptation before online adaptation begins. Additionally, we study the effect of *incremental context update* and why it is a key enabler for ACE's performance gain in Appendix §A.5.

| Method | GT Labels | Test-Normal | | Test-Challenge | | Average |
|---|---|---|---|---|---|---|
| | | TGC↑ | SGC↑ | TGC↑ | SGC↑ | |
| DeepSeek-V3.1 as Base LLM | | | | | | |
| ReAct | | 63.7 | 42.9 | 41.5 | 21.6 | 42.4 |
| Offline Adaptation | | | | | | |
| ReAct + ACE w/o Reflector or multi-epoch | ✓ | $70.8_{+7.1}$ | $55.4_{+12.5}$ | $55.9_{+14.4}$ | $38.1_{+17.5}$ | $55.1_{+12.7}$ |
| ReAct + ACE w/o multi-epoch | ✓ | $72.0_{+8.3}$ | $60.7_{+17.8}$ | $54.9_{+13.4}$ | $39.6_{+18.0}$ | $56.8_{+14.4}$ |
| ReAct + ACE | ✓ | $76.2_{+12.5}$ | $64.3_{+21.4}$ | $57.3_{+15.8}$ | $39.6_{+18.0}$ | $59.4_{+17.0}$ |
| Online Adaptation | | | | | | |
| ReAct + ACE | ✗ | $67.9_{+4.2}$ | $51.8_{+8.9}$ | $61.4_{+19.9}$ | $43.2_{+21.6}$ | $56.1_{+13.7}$ |
| ReAct + ACE + offline warmup | ✗ | $69.6_{+5.9}$ | $53.6_{+10.7}$ | $66.0_{+24.5}$ | $48.9_{+27.3}$ | $59.5_{+17.1}$ |

Table 3: **Ablation Studies on AppWorld.** We study how particular design choices of ACE (iterative refinement, multi-epoch adaptation, and offline warmup) could help high-quality context adaptation.

**Robustness to Reflection Quality**  ACE is robust to reflection quality: it remains effective with a much weaker Reflector and shows only modest additional gains from stronger reflectors, and it degrades gracefully under noisy/harmful reflections, staying above the base model except under fully adversarial updates every iteration. Full experiment results are in Appendix §A.4.

**Sensitivity to Hyperparameter Choice**  ACE's gains are stable across a wide range of reasonable hyperparameter settings (*e.g.,* Reflector refinement rounds, number of adaptation epochs, and grow-and-refine thresholds): performance changes are modest, and ACE consistently remains above the corresponding baselines. Full discussion and detailed results are reported in Appendix §A.6.

## 4.7 COST AND SPEED ANALYSIS

Due to its support for incremental, "delta" context updates and non-LLM-based context merging and de-duplication, ACE demonstrates particular advantages in reducing the cost (in terms of the number of rollouts or the amount of dollar cost for token ingestion/generation) and latency of adaptation.

As examples, on the offline adaptation of AppWorld, ACE achieves 82.3% reduction in adaptation latency and 75.1% reduction in the number of rollouts as compared to GEPA (Table 4(a)).  On

the online adaptation of FiNER, ACE achieves 91.5% reduction in adaptation latency and 83.6% reduction in token dollar cost for token ingestion/generation as compared to DC (Table 4(b)).

| Method | Latency (s)↓ | # Rollouts↓ |
|---|---|---|
| ReAct + GEPA | 53898 | 1434 |
| ReAct + ACE | 9517(-82.3%) | 357(-75.1%) |

(a) **Offline** (AppWorld).

| Method | Latency (s)↓ | Token Cost ($)↓ |
|---|---|---|
| DC (CU) | 65104 | 17.7 |
| ACE | 5503(-91.5%) | 2.9(-83.6%) |

(b) **Online** (FiNER).

Table 4: **Cost and Speed Analysis.** We measure the context adaptation latency, number of rollouts, and dollar costs of ACE against GEPA (offline) and DC (online).

**Fine-Grained Cost Analysis** We conduct a fine-grained cost analysis of ACE and GEPA (as a representative baseline). On AppWorld, ACE is substantially cheaper during offline adaptation, reducing input/output token usage by **80.8%/83.6%** vs. GEPA: ACE avoids GEPA's prompt-validation loop and replaces repeated full rewrites with localized delta updates. At evaluation time, while ACE may use more *raw* input tokens due to a richer playbook, this does not necessarily translate to higher *billed* serving cost because a large fraction of the context is reused by KV caching; we quantify this effect in the next paragraph. Full results, including a component-wise token breakdown for both methods, are in Appendix §A.3.

**KV Cache Reuse: Longer Context ≠ Higher Serving Cost** Although ACE produces longer contexts than methods such as GEPA, this does not translate to linearly higher inference cost or GPU memory usage. Modern serving infrastructures are increasingly optimized for long-context workloads through techniques such as the reuse (Gim et al., 2024; Yao et al., 2025), compression (Liu et al., 2024c;b), and offload (Lee et al., 2024; Li et al., 2025) of KV cache. These mechanisms allow frequently reused context segments to be cached locally or remotely, avoiding repetitive and expensive prefill operations. Ongoing advances in ML systems suggest that the amortized cost of handling long contexts is likely to decrease, making context-rich approaches like ACE increasingly practical in deployment. In our prompt-caching study with the OpenAI API (GPT-5.1), we find that ACE achieves *high cache reuse*: **91.8%** of input tokens are served from cache during evaluation stage, which reduces billed input-token cost by **82.6%** relative to counting raw context tokens.

## 5 DISCUSSION

**Implications for Online and Continual Learning** Online and continual learning are key research directions in machine learning for addressing issues like distribution shifts (Koh et al., 2021; Gulrajani & Lopez-Paz, 2021) and limited training data (Pan & Yang, 2010; Hutchinson et al., 2017; Zhuang et al., 2019). ACE offers a flexible and efficient alternative to conventional model finetuning, as adapting contexts is generally cheaper than updating model weights (Brown et al., 2020; Lester et al., 2021; Li & Liang, 2021; Hu et al., 2022). Moreover, because contexts are humaninterpretable, ACE enables *selective unlearning* (Cao & Yang, 2015; Bourtoule et al., 2021; Liu et al., 2025a), whether due to privacy or legal constraints (gdp, 2016; ccp, 2018), or when outdated or incorrect information is identified by domain experts. These are promising directions for future work, where ACE could play a central role in advancing continuous and responsible learning.

**Limitations and Challenges** A limitation of ACE is its reliance on a reasonably strong Reflector: if the Reflector fails to extract meaningful insights from generated traces or outcomes, the constructed context may become noisy or even harmful. In domain-specific tasks where no model can extract useful insights, the resulting context will naturally lack them. This dependency is similar to Dynamic Cheatsheet (Suzgun et al., 2025), where the quality of adaptation hinges on the underlying model's ability to curate memory. We also note that not all applications require rich or detailed contexts. Tasks like HotPotQA (Yang et al., 2018) often benefit more from concise, high-level instructions (*e.g.,* how to retrieve and synthesize evidence) than from long contexts. Similarly, games with fixed strategies such as Game of 24 (Suzgun et al., 2025) may only need a single reusable rule, rendering additional context redundant. Overall, ACE is most beneficial in settings that demand detailed domain knowledge, complex tool use, or environment-specific strategies that go beyond what is already embedded in model weights or simple system instructions.

ACKNOWLEDGEMENT

We thank the anonymous reviewers and area chair for their constructive feedback, which improved this paper. Qizheng Zhang is supported by NSF award CNS-2211384 and DARPA award TFAWI-HR00112520038. We also thank Lakshya A Agrawal, Xuekai Zhu, Yuhan Liu, Junchen Jiang, and Azalia Mirhoseini for helpful discussions.

ETHICS STATEMENT

This work does not raise specific ethical concerns. Our contributions focus on developing algorithms and system frameworks for effective context adaptation in large language models (LLMs). All experiments are conducted on publicly available benchmarks with open-source models, without involving human subjects, sensitive data, or privacy-related information. No potential conflicts of interest are present.

REPRODUCIBILITY STATEMENT

Our code is available at `github.com/ace-agent/ace`. We provide detailed descriptions of our experimental setup, including datasets, benchmarks, evaluation metrics, baselines, and hyper-parameter choices. Additional details, such as prompts for large language models and extended experimental settings, are included in the appendix. With this information, readers with reasonable computational resources should be able to reproduce our results.

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

# A  EXTENDED RESULTS

## A.1  GENERALIZATION ACROSS DIFFERENT LLMs

ACE is a model-agnostic framework that operates on execution traces and contextual deltas, and does not rely on any architectural or training-specific features of DeepSeek-V3.1 (the default backbone we use in the main text). We evaluated ACE with three additional models of varying size, cost, and capability: GPT-OSS-120B (Table 5 and 7), GPT-5.1 (Table 6 and 8), and Llama-3.3-70B-Instruct (Table 9). In each case, the Generator, Reflector, and Curator were all switched to the new model without changes to the algorithm.

**Analysis**   Across all four LLM families we tested (DeepSeek-V3.1, GPT-OSS-120B, GPT-5.1, and Llama-3.3-70B-Instruct), ACE consistently improves performance over both the base LLM/agent, GEPA, and other baselines, often by 5 to 12 points depending on the task and supervision setting. The relative gains of ACE remain stable even when switching to models that differ significantly in size, cost, and training recipe, and ACE delivers benefits with or without ground-truth labels, validating the robustness of its context adaptation mechanism. The online variant reliably achieves the strongest performance across all models.

We note that the magnitude of improvement can vary across model families: for example, Llama-3.3-70B-Instruct shows smaller gains compared to GPT-5.1 or GPT-OSS-120B. This is expected since ACE relies on the quality of intermediate reflections and calibrations, and smaller or weaker models naturally generate noisier feedback. Even in these cases, however, ACE remains beneficial, demonstrating that the framework is broadly applicable while still reflecting the inherent capability limits of the underlying LLM, as we discussed in the "Limitations and Challenges" paragraph in §5.

| Method | GT Labels | Test-Normal | | Test-Challenge | | Average |
|---|---|---|---|---|---|---|
| | | TGC↑ | SGC↑ | TGC↑ | SGC↑ | |
| GPT-OSS-120B as Base LLM | | | | | | |
| ReAct | | 54.8 | 33.9 | 34.5 | 15.1 | 34.6 |
| Offline Adaptation | | | | | | |
| ReAct + GEPA | ✓ | $56.0_{+1.2}$ | $33.9_{+0.0}$ | $40.1_{+5.6}$ | $20.9_{+5.8}$ | $37.7_{+3.1}$ |
| ReAct + ACE | ✓ | $\mathbf{61.3}_{+6.5}$ | $39.3_{+5.4}$ | $\mathbf{40.3}_{+5.8}$ | $20.9_{+5.8}$ | $\mathbf{40.5}_{+5.9}$ |
| ReAct + ACE | ✗ | $58.3_{+3.5}$ | $\mathbf{41.1}_{+7.2}$ | $39.6_{+5.1}$ | $18.7_{+3.6}$ | $39.4_{+4.8}$ |
| Online Adaptation | | | | | | |
| ReAct + DC (CU) | ✗ | $49.4_{-5.4}$ | $33.9_{+0.0}$ | $30.8_{-3.7}$ | $18.2_{+3.1}$ | $33.1_{-1.5}$ |
| ReAct + ACE | ✗ | $\mathbf{60.7}_{+5.9}$ | $\mathbf{44.6}_{+10.7}$ | $\mathbf{43.2}_{+8.7}$ | $\mathbf{20.1}_{+5.0}$ | $\mathbf{42.2}_{+7.6}$ |

Table 5: **Results on the AppWorld Agent Benchmark (GPT-OSS-120B as the Base LLM).** "GT labels" indicates whether ground-truth labels are available to the Reflector during adaptation. We evaluate the ACE framework against multiple baselines on top of the official ReAct implementation, both for offline and online context adaptation.

## A.2  BEYOND FINANCE: ADDITIONAL DOMAIN TASKS

We evaluate ACE on two additional domain tasks from StreamBench (Wu et al., 2024) under the non-streaming setting (*i.e.,* offline adaptation): DDXPlus (Fansi Tchango et al., 2022) for medical reasoning (Table 10) and BIRD-SQL (Li et al., 2023) for Text-to-SQL generation (Table 11). For ACE, we perform offline adaptation using 1000 randomly sampled training examples. For GEPA, we use the same 1000 examples for training, and reserve a separate validation set of 500 examples for BIRD-SQL or 372 examples for DDXPlus (StreamBench provides 1372 train/val examples for DDXPlus in total). All other settings follow the main-text configuration unless stated otherwise.

**Analysis**   On DDXPlus, ACE substantially improves over the base LLM, rising from 75.2 to 90.2 accuracy (**+15.0**). In contrast, GEPA yields a much smaller gain (76.4, **+1.2**). This suggests that ACE's test-time evolving context transfers well to multi-step, domain-heavy diagnostic reasoning.

| Method | GT Labels | Test-Normal | | Average |
| --- | --- | --- | --- | --- |
| | | TGC↑ | SGC↑ | |
| GPT-5.1 as Base LLM | | | | |
| ReAct | | 61.9 | 46.4 | 54.2 |
| Offline Adaptation | | | | |
| ReAct + GEPA | ✓ | $64.3_{+2.4}$ | $48.2_{+1.8}$ | $56.2_{+2.0}$ |
| ReAct + ACE | ✓ | $66.7_{+4.8}$ | $53.6_{+7.2}$ | $60.2_{+6.0}$ |
| ReAct + ACE | ✗ | $\mathbf{67.3}_{+5.4}$ | $\mathbf{55.4}_{+9.0}$ | $\mathbf{61.3}_{+7.1}$ |
| Online Adaptation | | | | |
| ReAct + DC (CU) | ✗ | $62.5_{+0.6}$ | $55.4_{+9.0}$ | $58.9_{+4.7}$ |
| ReAct + ACE | ✗ | $\mathbf{72.6}_{+10.7}$ | $\mathbf{58.9}_{+12.5}$ | $\mathbf{65.8}_{+11.6}$ |

Table 6: **Results on the AppWorld Agent Benchmark (GPT-5.1 as the Base LLM).** "GT labels" indicates whether ground-truth labels are available to the Reflector during adaptation. We evaluate the ACE framework against multiple baselines on top of the official `ReAct` implementation, both for offline and online context adaptation.

| Method | GT Labels | FiNER (Acc↑) | Formula (Acc↑) | Average |
| --- | --- | --- | --- | --- |
| GPT-OSS-120B as Base LLM | | | | |
| Base LLM | | 66.6 | 71.5 | 69.1 |
| Offline Adaptation | | | | |
| GEPA | ✓ | $67.9_{+1.3}$ | $71.5_{+0.0}$ | $69.7_{+0.6}$ |
| ACE | ✓ | $\mathbf{73.8}_{+7.2}$ | $\mathbf{88.5}_{+17.0}$ | $\mathbf{81.2}_{+12.1}$ |
| ACE | ✗ | $69.7_{+3.1}$ | $84.0_{+12.5}$ | $76.9_{+7.8}$ |
| Online Adaptation | | | | |
| DC | ✗ | $55.8_{-10.8}$ | $66.0_{-5.5}$ | $60.9_{-8.2}$ |
| ACE | ✗ | $\mathbf{70.5}_{+3.9}$ | $\mathbf{85.0}_{+13.5}$ | $\mathbf{77.8}_{+8.7}$ |

Table 7: **Results on Financial Analysis Benchmark (GPT-OSS-120B as the Base LLM).** "GT labels" indicates whether ground-truth labels are available to the Reflector during adaptation.

| Method | GT Labels | FiNER (Acc↑) | Formula (Acc↑) | Average |
| --- | --- | --- | --- | --- |
| GPT-5.1 as Base LLM | | | | |
| Base LLM | | 73.5 | 73.0 | 73.3 |
| Offline Adaptation | | | | |
| GEPA | ✓ | $74.5_{+1.0}$ | $73.0_{+0.0}$ | $73.8_{+0.5}$ |
| ACE | ✓ | $\mathbf{81.0}_{+7.5}$ | $\mathbf{84.5}_{+11.5}$ | $\mathbf{82.8}_{+9.5}$ |
| ACE | ✗ | $78.2_{+4.7}$ | $76.5_{+3.5}$ | $77.4_{+4.1}$ |
| Online Adaptation | | | | |
| DC | ✗ | $70.0_{-3.5}$ | $69.0_{-4.0}$ | $69.5_{-3.8}$ |
| ACE | ✗ | $\mathbf{75.2}_{+1.7}$ | $\mathbf{79.0}_{+6.0}$ | $\mathbf{77.1}_{+3.8}$ |

Table 8: **Results on Financial Analysis Benchmark (GPT-5.1 as the Base LLM).** "GT labels" indicates whether ground-truth labels are available to the Reflector during adaptation.

On BIRD-SQL, ACE also improves consistently over the base model on all splits, achieving better overall average (52.9, **+5.1**). The gains are driven mainly by the Simple subset (53.5, **+7.1**). While GEPA yields larger gains on Moderate and Challenging, ACE still improves over the base model on both splits. Overall, these results indicate that ACE generalizes beyond finance to both knowledge-intensive reasoning and structured code generation tasks.

| Method | GT Labels | FiNER (Acc↑) |
|---|---|---|
| Llama-3.3-70B-Instruct as Base LLM | | |
| Base LLM | | 62.5 |
| Offline Adaptation | | |
| GEPA | ✓ | $59.41_{-3.09}$ |
| ACE | ✓ | $\mathbf{64.9}_{+2.4}$ |
| ACE | ✗ | $64.2_{+1.7}$ |
| Online Adaptation | | |
| DC | ✗ | $59.0_{-3.5}$ |
| ACE | ✗ | $\mathbf{63.6}_{+1.1}$ |

Table 9: **Results on Financial Analysis Benchmark (Llama-3.3-70B-Instruct as the Base LLM).** "GT labels" indicates whether ground-truth labels are available to the Reflector during adaptation.

| Method | Accuracy (↑) |
|---|---|
| DeepSeek-V3.1 as Base LLM | |
| Base LLM | 75.2 |
| Offline Adaptation | |
| GEPA | $76.4_{+1.2}$ |
| ACE | $\mathbf{90.2}_{+15.0}$ |

Table 10: **Results on Medical Reasoning Benchmark (DeepSeek-V3.1-671B as the Base LLM).** We use DDXPlus from StreamBench.

| Method | Simple (↑) | Moderate (↑) | Challenging (↑) | Average |
|---|---|---|---|---|
| DeepSeek-V3.1 as Base LLM | | | | |
| Base LLM | 46.4 | 48.2 | 55.1 | 47.8 |
| Offline Adaptation | | | | |
| GEPA | $51.6_{+5.2}$ | $\mathbf{51.9}_{+3.7}$ | $\mathbf{57.2}_{+2.1}$ | $52.2_{+4.4}$ |
| ACE | $\mathbf{53.5}_{+7.1}$ | $50.7_{+2.5}$ | $56.6_{+1.5}$ | $\mathbf{52.9}_{+5.1}$ |

Table 11: **Results on Text-to-SQL Benchmark (DeepSeek-V3.1-671B as the Base LLM).** We use BIRD-SQL from StreamBench.

## A.3 FINE-GRAINED COST ANALYSIS

We perform a fine-grained cost analysis of ACE and GEPA for both the *adaptation* and *evaluation* stages (offline adaptation setting), using AppWorld as a representative application. For ACE, we run offline adaptation with 1 epoch and 1 reflector refinement round. While increasing the number of epochs or refinement rounds will increase cost, the qualitative trends below should remain: ACE avoids expensive validation-time re-evaluation and performs localized updates rather than repeated full rewrites. For GEPA, we use the official DSPy implementation (DSPy, b) with `auto="heavy"` to maximize optimization strength.

**Adaptation Stage**   Across adaptation (Table 12 and Table 13), ACE reduces input-token usage by 80.8% relative to GEPA (204.1M → 39.3M) and output-token usage by 83.6% (1.87M → 0.31M). This gap is primarily driven by (1) GEPA's prompt-validation loop, which repeatedly evaluates candidate prompts on a held-out validation set (57 queries), incurring substantial additional LLM calls and validation tokens, and (2) ACE's incremental context updates, which replace full prompt rewrites with localized Generator-Reflector-Curator updates.

**Evaluation Stage**   At evaluation time (Table 14 and Table 15), ACE uses more *raw* input tokens per query than GEPA due to its richer, more actionable playbook. However, output-token usage

is similar (115.0 vs. 101.8), and the number of rollouts is comparable across methods. Moreover, under modern prompt/KV-caching infrastructures, the additional input tokens can be largely amortized: using OpenAI's default prompt caching, 91.8% of ACE's input tokens are served from cache, resulting in an 82.6% reduction in billed input-token cost (see §4.7).

| Method | Total # input tokens | Total # output tokens | Total # rollouts | Total # queries |
|---|---|---|---|---|
| GEPA (prompt generation) | 65,005,192 | 1,266,090 | 429 | 90 |
| GEPA (prompt validation) | 139,070,904 | 604,098 | 1,026 | 57 |
| GEPA (total) | 204,076,096 | 1,870,188 | 1,455 | 147 |
| ACE (Generator) | 31,012,122 | 198,847 | 1,790 | 90 |
| ACE (Reflector) | 4,685,840 | 70,487 | 161 | 90 |
| ACE (Curator) | 3,552,963 | 37,794 | 124 | 90 |
| ACE (total) | 39,250,925 (-80.8%) | 307,128 (-83.6%) | 2,075 (+42.6%) | 90 (-38.8%) |

Table 12: **Adaptation-Stage Aggregate Cost Statistics on AppWorld.** The percentages compare ACE (total) against GEPA (total).

| Method | Avg # input / rollout | Avg # output / rollout | Avg # input / query | Avg # output / query | Total # rollouts | Total # queries |
|---|---|---|---|---|---|---|
| GEPA (prompt generation) | 151,600.4 | 2,951.4 | 722,279.9 | 14,067.7 | 429 | 90 |
| GEPA (prompt validation) | 135,550.8 | 589.2 | 2,439,840.4 | 10,600.0 | 1,026 | 57 |
| GEPA (total) | 140,291.8 | 1,285.4 | 1,387,538.4 | 12,721.7 | 1,455 | 147 |
| ACE (Generator) | 17,326.6 | 111.1 | 344,579.1 | 2,209.4 | 1,790 | 90 |
| ACE (Reflector) | 29,112.4 | 437.3 | 52,064.9 | 783.2 | 161 | 90 |
| ACE (Curator) | 28,667.4 | 304.8 | 39,477.4 | 420.0 | 124 | 90 |
| ACE (total) | 18,914.9 (-86.5%) | 148.0 (-88.5%) | 436,121.4 (-68.6%) | 3,412.5 (-73.2%) | 2,075 (+42.6%) | 90 (-38.8%) |

Table 13: **Adaptation-Stage Average Cost Statistics on AppWorld.** The percentages compare ACE (total) against GEPA (total).

| Method | Total # input tokens | Total # output tokens | Total # rollouts | Total # eval queries |
|---|---|---|---|---|
| Base ReAct | 27,460,411 | 289,802 | 2,430 | 160 |
| GEPA | 26,960,675 | 251,442 | 2,470 | 160 |
| ACE | 58,623,267 (+117.4%) | 270,652 (+7.6%) | 2,354 (-4.7%) | 160 (+0.0%) |

Table 14: **Evaluation-Stage Aggregate Cost Statistics on AppWorld.** The percentages compare ACE against GEPA.

| Method | Avg # input / rollout | Avg # output / rollout | Avg # input / query | Avg # output / query | Total # rollouts | Total # eval queries |
|---|---|---|---|---|---|---|
| Base ReAct | 11,298.5 | 119.3 | 171,627.6 | 1,811.3 | 2,430 | 160 |
| GEPA | 10,918.5 | 101.8 | 168,504.2 | 1,571.5 | 2,470 | 160 |
| ACE | 24,912.4 (+128.2%) | 115.0 (+13.0%) | 366,395.4 (+117.4%) | 1,691.6 (+7.6%) | 2,354 (-4.7%) | 160 (+0.0%) |

Table 15: **Evaluation-Stage Average Cost Statistics on AppWorld.** The percentages compare ACE against GEPA.

## A.4 ROBUSTNESS TO REFLECTION QUALITY

We conduct two analyses to evaluate ACE's sensitivity to reflection quality. Unless otherwise noted, experiments are run on FiNER with ACE offline adaptation.

**Using Weaker Reflector Models**    To test whether ACE requires a strong reflector, we vary the Reflector across three models with substantially different capability: GPT-OSS-120B, DeepSeek-V3.1-671B, and GPT-5.1 (Table 16). Across all choices, ACE consistently improves over the base LLM, including when the Reflector is much weaker. While stronger reflectors yield larger gains, the method remains effective across a wide range of reflector strengths.

**Robustness to Noisy or Harmful Reflector Feedback**    We further stress-test ACE with actively harmful reflector outputs by injecting adversarial or conflicting bullets: we invoke a "harmful" reflector that is explicitly instructed to inject harmful reflection once every $X$ adaptation steps, where

| Method | Generator LLM | Reflector LLM | Curator LLM | Accuracy (↑) |
|---|---|---|---|---|
| Base LLM | DeepSeek-V3.1 | - | - | 70.7 |
| ACE | DeepSeek-V3.1 | GPT-OSS-120B | DeepSeek-V3.1 | 76.6 (+5.9) |
| ACE | DeepSeek-V3.1 | DeepSeek-V3.1 | DeepSeek-V3.1 | 78.3 (+7.6) |
| ACE | DeepSeek-V3.1 | GPT-5.1 | DeepSeek-V3.1 | 78.5 (+7.8) |

Table 16: **Weaker Reflector Models on FiNER.** We vary the Reflector while keeping the Generator/Curator fixed. Parentheses report deltas vs. the base LLM.

| Harmful Reflector Frequency (every $X$ iters) | Accuracy (↑) |
|---|---|
| base LLM | 70.7 |
| 1 | 66.7 (-4.0) |
| 5 | 76.1 (+5.4) |
| 10 | 77.0 (+6.3) |
| 25 | 77.8 (+7.1) |
| 50 | 78.2 (+7.5) |
| 100 | 78.2 (+7.5) |
| No harmful reflector | 78.3 (+7.6) |

Table 17: **Robustness to Noisy or Harmful Reflector Feedback on FiNER.** We invoke a harmful reflector once every $X$ adaptation steps. Parentheses report deltas vs. the base LLM.

larger $X$ means less frequent corruption (Table 17). ACE is robust to moderate noise levels: performance degrades gradually as corruption becomes more frequent and stays above the base LLM except in the extreme case of injecting harmful updates every iteration. This suggests that ACE's update mechanism tolerates substantial noise in the reflection stream, with failures emerging only under intentionally adversarial conditions.

**Takeaway and Mitigation** ACE is not highly sensitive to reflector quality: (1) weaker reflectors still provide substantial gains, (2) moderate noise or conflicting updates are largely tolerated, and (3) performance drops below the base model only under sustained adversarial corruption. In practice, ACE's bullet-point analyzer ("grow-and-refine"; §3.2), which merges and deduplicates semantically similar bullets and can filter entries flagged as potentially harmful via metadata, serves as a first line of defense against context noise. Additional safeguards (*e.g.,* contradiction detection, prompting the Curator to prioritize high-confidence updates, or periodic pruning of outdated entries) are compatible extensions that could further improve playbook compactness and consistency (Zhou et al., 2024).

## A.5 ABLATION ON INCREMENTAL CONTEXT UPDATE

In this ablation, we run offline context adaptation on AppWorld using ACE with and without incremental context updates, and evaluate on the test-normal split with DeepSeek-V3.1. We find that incremental updates are critical: by preserving useful information that would otherwise be lost to context collapse, they account for a large share of ACE 's gains.

| Method | Test-Normal TGC (↑) | Test-Normal SGC (↑) | Average |
|---|---|---|---|
| DeepSeek-V3.1 as Base LLM | | | |
| ReAct | 63.7 | 42.9 | 53.3 |
| Offline Adaptation | | | |
| ReAct + ACE (no incremental update) | $67.3_{+3.6}$ | $46.4_{+3.5}$ | $56.9_{+3.6}$ |
| ReAct + ACE (with incremental update) | $\mathbf{76.2}_{+12.5}$ | $\mathbf{64.3}_{+21.4}$ | $\mathbf{70.3}_{+17.0}$ |

Table 18: **Ablation on Incremental Context Updates (AppWorld, DeepSeek-V3.1).** We run offline context adaptation with ACE with/without incremental updates and evaluate on test-normal. Improvements are relative to ReAct.

A.6   SENSITIVITY ANALYSIS ON HYPERPARAMETER CHOICE

**Reflection Iterations**   This parameter trades off (1) extracting enough high-quality insights from the inference traces and (2) avoiding "overthinking" that introduces noisy or unnecessary updates. Empirically, 5 rounds offers a good balance. On AppWorld, 1 round under-extracts useful strategy fragments and leaves clear headroom, while too many rounds (*e.g.,* 10) can degrade performance.

| # Reflection Iterations | Test-Normal TGC (↑) | Test-Normal SGC (↑) | Average |
|---|---|---|---|
| DeepSeek-V3.1 as Base LLM | | | |
| N/A (without ACE) | 63.7 | 42.9 | 53.3 |
| Offline Adaptation | | | |
| 1 | $69.0_{+5.3}$ | $53.6_{+10.7}$ | $61.3_{+8.0}$ |
| 3 | $74.4_{+10.7}$ | $57.1_{+14.2}$ | $65.8_{+12.5}$ |
| 5 | $72.6_{+8.9}$ | $62.5_{+19.6}$ | $67.6_{+14.3}$ |
| 10 | $71.4_{+7.7}$ | $58.9_{+16.0}$ | $65.2_{+11.9}$ |

Table 19: **Effect of Reflection Iterations (AppWorld, DeepSeek-V3.1).**   We report test-normal results under offline context adaptation. Improvements are relative to running ReAct without ACE.

**Deduplication Threshold**   This controls how aggressively newly extracted insights are merged with existing entries. On FiNER, performance changes only mildly across the tested range, suggesting ACE is robust to moderate variation in dedup aggressiveness.

| Deduplication Threshold | FiNER (Acc ↑) |
|---|---|
| DeepSeek-V3.1 as Base LLM | |
| N/A (without ACE) | 70.7 |
| Offline Adaptation | |
| 50% | $77.0_{+6.3}$ |
| 70% | $73.9_{+3.2}$ |
| 90% | $78.6_{+7.9}$ |

Table 20: **Effect of Deduplication Threshold (FiNER, DeepSeek-V3.1).**

**Pruning Trigger (Maximum Context Length)**   This threshold determines when ACE merges and prunes older or low-utility entries to prevent unbounded growth. It balances (1) retaining enough accumulated context to improve learning from experience and (2) keeping the context compact to reduce cost and limit noise. On FiNER, performance is stable from 10K to 100K tokens, indicating ACE does not require finely tuned length thresholds; pruning mainly removes stale or harmful fragments while preserving core reusable strategies.

| Max Context Length | FiNER (Acc ↑) |
|---|---|
| DeepSeek-V3.1 as Base LLM | |
| N/A (without ACE) | 70.7 |
| Offline Adaptation | |
| 10K | $78.6_{+7.9}$ |
| 50K | $78.4_{+7.7}$ |
| 100K | $78.3_{+7.6}$ |

Table 21: **Effect of Pruning Trigger (FiNER, DeepSeek-V3.1).**

Overall, ACE is not highly sensitive to these hyperparameters: reasonable choices (*e.g.,* 3-5 reflection rounds, 50-90% dedup threshold, and 10K-100K pruning triggers) consistently yield strong performance.

## B    EXTENDED RELATED WORK

### B.1    AGENT MEMORY

A growing body of work explores how agents can accumulate experience from past trajectories and leverage external (often non-parametric) memory to guide future actions. AgentFly (Zhou et al., 2025) presents an extensible framework where memory evolves continuously as agents solve tasks, enabling scalable reinforcement learning and long-horizon reasoning across diverse environments. AWM (Agent Workflow Memory) (Wang et al., 2025b) induces reusable *workflows*, *i.e.,* structured routines distilled from past trajectories, and selectively injects them into memory to improve efficiency and generalization in web navigation benchmarks. A-MEM (Xu et al., 2025) introduces a dynamically organized memory system inspired by the Zettelkasten method: each stored memory is annotated with structured attributes (*e.g.,* tags, keywords, contextual descriptions) and automatically linked to relevant past entries, while existing entries are updated to integrate new knowledge, yielding adaptive and context-aware retrieval. Agentic Plan Caching (Zhang et al., 2025c) instead focuses on cost efficiency by extracting reusable plan templates from agent trajectories and caching them for fast execution at test time.

Together, these works demonstrate the value of external memory for improving adaptability, efficiency, and generalization in LLM agents. Our work differs by tackling the broader challenge of *context adaptation*, which spans not only agent memory but also system prompts, factual evidence, and other inputs underpinning AI systems. We further highlight two fundamental limitations of existing adaptation methods: *brevity bias* and *context collapse*; and show that addressing them is essential for robustness, reliability, and scalability beyond raw task performance. Accordingly, our evaluation considers not only accuracy but also cost, latency, and scalability.

## C    EXTENDED DISCUSSIONS

### C.1    ACE VS. GEPA

**Scope and Objective** Both GEPA and ACE improve model or agent behavior by adapting *context* at test time (rather than updating model weights), but they are designed for different forms of adaptation. GEPA treats adaptation as *prompt evolution*: it iteratively proposes and selects improved instruction prompts using rollout trajectories and reflective feedback, aiming to maximize a task evaluator under a rollout budget. ACE targets settings where performance depends on accumulating and preserving *many granular, reusable insights* over long horizons. For multi-turn agents (*e.g.,* AppWorld), the model must retain step-by-step procedures and tool-use rules across an interaction. For domain-specific and knowledge-intensive benchmarks (*e.g.,* FiNER, Formula), accuracy depends on keeping many specific rules, edge cases, and domain concepts that are difficult to compress into a single instruction prompt without losing detail.

**Update Mechanism and Representation** GEPA updates context by generating and selecting *new prompt variants* in an evolutionary loop, where each candidate is a full prompt optimized end-to-end. ACE instead represents context as a structured, itemized Playbook and applies *incremental delta updates*: the Curator writes only the new insight, and we merge it into the Playbook with simple deterministic logic. This avoids repeated full-prompt rewrites, helps keep earlier rules stable over long runs, and enables fine-grained bookkeeping (*e.g.,* de-duplication, targeted refinement, and tracking which entries helped or harmed accuracy).

### C.2    ACE VS. DYNAMIC CHEATSHEET (DC)

**Scope and Objective** Both DC and ACE collect reusable insights at test time, but they are aimed at different settings. DC is mainly evaluated on single-turn reasoning benchmarks (*e.g.,* AIME, Game-of-24, GPQA) where each query is independent. In this regime, improvements often come from saving short, reusable heuristics and executable artifacts (*e.g.,* code snippets) that help on later problems. ACE focuses on settings where the details and high-fidelity guidenace need to stick around. For multi-turn agents (*e.g.,* AppWorld), the model must remember step-by-step procedures and tool-use rules across an interaction. For domain-specific and knowledge-intensive benchmarks

(*e.g.,* FiNER, Formula), accuracy depends on keeping many specific rules, edge cases, and domain concepts, which are hard to compress without losing information.

**Update Mechanism** DC updates its memory by rewriting the cheatsheet as a whole, either by re-generating the full cheatsheet each step (DC-CU) or by writing a new summary from retrieved examples (DC-RS). With repeated full rewrites, the model tends to shorten and compress what was written before. This can cause context collapse, which means that useful domain-specific details get dropped over time or disappear suddenly (§2.2). ACE avoids full rewrites by using incremental delta updates. The Curator only writes the new insight, and we merge it into a structured and itemized Playbook with simple deterministic logic. This keeps earlier rules stable over long runs and also makes it easier to track which items helped, remove duplicates, and refine entries. In our ablations (§4.6), removing delta updates leads to a large drop on AppWorld (-11.7% TGC and -27.8% SGC on test-normal), showing that delta updates are a core part of our method.

# D    APPWORLD LEADERBOARD SNAPSHOT (09/2025)

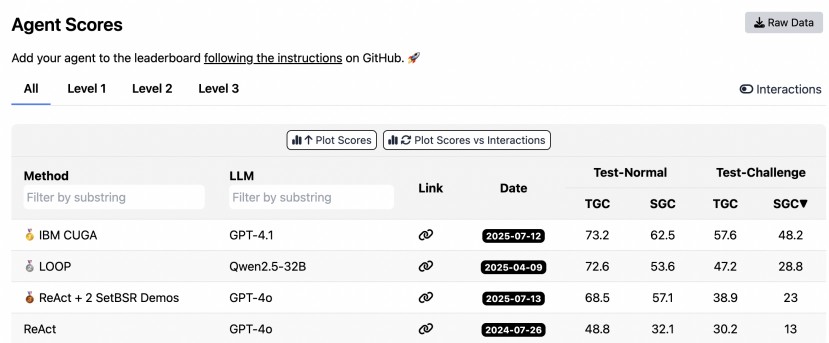

Figure 5: The AppWorld leaderboard as accessed on 09/2025.

# E    THE USE OF LARGE LANGUAGE MODELS (LLMS)

This work focuses on developing algorithms and system frameworks for effective context adaptation in large language models (LLMs). Accordingly, our experiments employ LLMs for the empirical evaluation of the proposed methods. For paper preparation, we used LLMs only to polish writing (*e.g.,* correcting grammatical errors), and not to generate new text from scratch.

# F  PROMPTS

I am your supervisor and you are a super intelligent AI Assistant whose job is to achieve my day-to-day tasks completely autonomously.

To do this, you will need to interact with app/s (e.g., spotify, venmo etc) using their associated APIs on my behalf. For this you will undertake a *multi-step conversation* using a python REPL environment. That is, you will write the python code and the environment will execute it and show you the result, based on which, you will write python code for the next step and so on, until you've achieved the goal. This environment will let you interact with app/s using their associated APIs on my behalf.

Here are three key APIs that you need to know to get more information

```python
# To get a list of apps that are available to you.
print(apis.api_docs.show_app_descriptions())

# To get the list of apis under any app listed above, e.g. spotify
print(apis.api_docs.show_api_descriptions(app_name='spotify'))

# To get the specification of a particular api, e.g. spotify app's login api
print(apis.api_docs.show_api_doc(app_name='spotify', api_name='login'))
```

Each code execution will produce an output that you can use in subsequent calls. Using these APIs, you can now generate code, that I will execute, to solve the task.

Let's start with the task

[3 shot example]

**Key instructions**:

1. Make sure to end code blocks with ``` followed by a newline().

2. Remember you can use the variables in your code in subsequent code blocks.

3. Remember that the email addresses, access tokens and variables (e.g. spotify_password) in the example above are not valid anymore.

4. You can use the "supervisor" app to get information about my accounts and use the "phone" app to get information about friends and family.

5. Always look at API specifications (using `apis.api_docs.show_api_doc`) before calling an API.

6. Write small chunks of code and only one chunk of code in every step. Make sure everything is working correctly before making any irreversible change.

7. Many APIs return items in "pages". Make sure to run through all the pages by looping over page_index.

8. Once you have completed the task, make sure to call `apis.supervisor.complete_task()`. If the task asked for some information, return it as the answer argument, i.e. call `apis.supervisor.complete_task(answer=<answer>)`. Many tasks do not require an answer, so in those cases, just call `apis.supervisor.complete_task()` i.e. do not pass any argument.

Using these APIs, generate code to solve the actual task:

My name is: {{ main_user.first_name }} {{ main_user.last_name }}. My personal email is {{ main_user.email }} and phone number is {{ main_user.phone_number }}.

Task: {{ input_str }}

Figure 6: ICL-baseline Generator prompt on AppWorld

I am your supervisor and you are a super intelligent AI Assistant whose job is to achieve my day-to-day tasks completely autonomously. You will be given a cheatsheet containing relevant strategies, patterns, and examples from similar problems to apply and solve the current task.

To do this, you will need to interact with app/s (e.g., spotify, venmo etc) using their associated APIs on my behalf. For this you will undertake a *multi-step conversation* using a python REPL environment. That is, you will write the python code and the environment will execute it and show you the result, based on which, you will write python code for the next step and so on, until you've achieved the goal. This environment will let you interact with app/s using their associated APIs on my behalf.

Here are three key APIs that you need to know to get more information

```python
# To get a list of apps that are available to you.
print(apis.api_docs.show_app_descriptions())

# To get the list of apis under any app listed above, e.g. spotify
print(apis.api_docs.show_api_descriptions(app_name='spotify'))

# To get the specification of a particular api, e.g. spotify app's login api
print(apis.api_docs.show_api_doc(app_name='spotify', api_name='login'))
```

Each code execution will produce an output that you can use in subsequent calls. Using these APIs, you can now generate code, that I will execute, to solve the task.

CHEATSHEET: ''' {{ cheat_sheet }} '''

---

### 1. ANALYSIS & STRATEGY

- Carefully analyze both the question and cheatsheet before starting
- Search for and identify any applicable patterns, strategies, or examples within the cheatsheet
- Create a structured approach to solving the problem at hand
- Review and document any limitations in the provided reference materials

### 2. SOLUTION DEVELOPMENT

- Present your solution using clear, logical steps that others can follow and review
- Explain your reasoning and methodology before presenting final conclusions
- Provide detailed explanations for each step of the process
- Check and verify all assumptions and intermediate calculations

### 3. PROGRAMMING TASKS

When coding is required: - Write clean, efficient Python code - Follow the strict code formatting and execution protocol (always use the Python code formatting block; furthermore, after the code block, always explicitly request execution by appending: "EXECUTE CODE!"): `python   # Your code here EXECUTE CODE!`

- All required imports and dependencies should be clearly declared at the top of your code
- Include clear inline comments to explain any complex programming logic
- Perform result validation after executing your code
- Apply optimization techniques from the cheatsheet when applicable
- The code should be completely self-contained without external file dependencies–it should be ready to be executed right away
- Do not include any placeholders, system-specific paths, or hard-coded local paths
- Feel free to use standard and widely-used pip packages
- Opt for alternative methods if errors persist during execution
- Exclude local paths and engine-specific settings (e.g., avoid configurations like chess.engine.SimpleEngine.popen_uci("/usr/bin/stockfish"))

Let's start with the task

[3 shot example]

---

**Key instructions**: (1) Make sure to end code blocks with ``` followed by a newline().

2. Remember you can use the variables in your code in subsequent code blocks.

3. Remember that the email addresses, access tokens and variables (e.g. spotify_password) in the example above are not valid anymore.

4. You can use the "supervisor" app to get information about my accounts and use the "phone" app to get information about friends and family.

5. Always look at API specifications (using `apis.api_docs.show_api_doc`) before calling an API.

6. Write small chunks of code and only one chunk of code in every step. Make sure everything is working correctly before making any irreversible change.

7. Many APIs return items in "pages". Make sure to run through all the pages by looping over `page_index`.

8. Once you have completed the task, make sure to call `apis.supervisor.complete_task()`. If the task asked for some information, return it as the answer argument, i.e. call `apis.supervisor.complete_task(answer=<answer>)`. Many tasks do not require an answer, so in those cases, just call `apis.supervisor.complete_task()` i.e. do not pass any argument.

Using these APIs, generate code to solve the actual task:

My name is: {{ main_user.first_name }} {{ main_user.last_name }}. My personal email is {{ main_user.email }} and phone number is {{ main_user.phone_number }}. Task: {{ input_str }}

Figure 7: Dynamic Cheatsheet Generator prompt on AppWorld

I am your supervisor and you are a super intelligent AI Assistant whose job is to achieve my day-to-day tasks completely autonomously.

To do this, you will need to interact with app/s (e.g., spotify, venmo etc) using their associated APIs on my behalf. For this you will undertake a *multi-step conversation* using a python REPL environment. That is, you will write the python code and the environment will execute it and show you the result, based on which, you will write python code for the next step and so on, until you've achieved the goal. This environment will let you interact with app/s using their associated APIs on my behalf.

Here are three key APIs that you need to know to get more information:

```python
# To get a list of apps that are available to you.
print(apis.api_docs.show_app_descriptions())

# To get the list of apis under any app listed above, e.g. spotify
print(apis.api_docs.show_api_descriptions(app_name='spotify'))

# To get the specification of a particular api, e.g. spotify app's login api
print(apis.api_docs.show_api_doc(app_name='spotify', api_name='login'))
```

Each code execution will produce an output that you can use in subsequent calls. Using these APIs, you can now generate code, that I will execute, to solve the task.

**Key Instructions:**

1. Always end code blocks with ``` followed by a newline().

2. Remember you can use variables in your code in subsequent code blocks.

3. Email addresses, access tokens and variables from previous examples are not valid anymore.

4. Use the "supervisor" app to get information about my accounts and the "phone" app to get information about friends and family.

5. Always look at API specifications (using `apis.api_docs.show_api_doc`) before calling an API.

6. Write small chunks of code and only one chunk of code in every step. Make sure everything is working correctly before making any irreversible changes.

7. Many APIs return items in "pages". Make sure to run through all the pages by looping over `page_index`.

8. Once you have completed the task, call `apis.supervisor.complete_task()`. If the task asked for information, return it as the answer argument: `apis.supervisor.complete_task(answer=<answer>)`. For tasks without required answers, just call `apis.supervisor.complete_task()` without arguments.

**Domain-Specific Strategy for Bill Splitting Tasks:** When splitting bills among roommates, remember to: - First identify roommates using phone app's search_contacts with "roommate" relationship query - Access bill receipts in file system under "/home/[username]/bills/" directory structure - Calculate equal shares by dividing total amount by (number of roommates + 1) including yourself - Use Venmo's create_payment_request API with roommates' email addresses - Ensure payment requests are only sent to actual roommates (not coworkers or other contacts) - Verify that all roommates have the same home address in their contact information - Use the description "I paid for cable bill." for payment requests

**Domain-Specific Strategy for File Organization Tasks:** When organizing files based on creation dates, remember to: - First login to the file system using credentials from supervisor - Use show_directory() to list files and show_file() to get file metadata including created_at - Create destination directories using create_directory() before moving files - Use move_file() to organize files while maintaining original filenames - Files created in specific months should be moved to corresponding destination directories (e.g., March → Rome, April → Santorini, others → Berlin)

**Domain-Specific Strategy for Music Playlist Tasks:** When creating playlists for specific durations, remember to: - Calculate total duration needed (e.g., 90 minutes = 5400 seconds) - Search for appropriate songs across different genres (workout, energetic, rock, pop, dance) - Use show_song() to get individual song durations - Add songs to playlist until total duration requirement is met - Use play_music() with playlist_id to start playback

**Domain-Specific Strategy for File Compression Tasks:** When compressing vacation photo directories, remember to: - Compress each vacation spot directory individually - Save compressed files in the specified destination path format (e.g., "~/photographs/vacations/.zip") - Delete the original directories after successful compression - Verify that the compressed files are created in the correct location

**Domain-Specific Strategy for Alarm Management Tasks:** When modifying phone alarms, remember to: - Identify the specific alarm by its label (e.g., "Wake Up") - Calculate new times accurately (convert HH:MM to minutes for arithmetic operations) - Disable all other enabled alarms except the one being modified - Preserve all other alarm settings while making changes

**Domain-Specific Strategy for Message Management Tasks:** When handling text/voice messages, remember to: - Use search functions to find specific messages by phone number or content - Handle pagination to ensure all relevant messages are processed - Delete messages using their specific message IDs - Verify deletion by checking that no messages remain

Let's start with the task:

Figure 8: GEPA prompt on AppWorld

I am your supervisor and you are a super intelligent AI Assistant whose job is to achieve my day-to-day tasks completely autonomously.

To do this, you will need to interact with app/s (e.g., spotify, venmo etc) using their associated APIs on my behalf. For this you will undertake a *multi-step conversation* using a python REPL environment. That is, you will write the python code and the environment will execute it and show you the result, based on which, you will write python code for the next step and so on, until you've achieved the goal. This environment will let you interact with app/s using their associated APIs on my behalf.

Here are three key APIs that you need to know to get more information

```python
# To get a list of apps that are available to you.
print(apis.api_docs.show_app_descriptions())

# To get the list of apis under any app listed above, e.g. spotify
print(apis.api_docs.show_api_descriptions(app_name='spotify'))

# To get the specification of a particular api, e.g. spotify app's login api
print(apis.api_docs.show_api_doc(app_name='spotify', api_name='login'))
```

Each code execution will produce an output that you can use in subsequent calls. Using these APIs, you can now generate code, that I will execute, to solve the task.

You are also provided with a curated cheatsheet of strategies, API-specific information, common mistakes, and proven solutions to help you solve the task effectively.

**ACE Playbook**: - Read the **Playbook** first, then execute the task by explicitly leveraging each relevant section:

**PLAYBOOK_BEGIN**

{{ playbook }}

**PLAYBOOK_END**

Let's start with the task

[3 shot example]

---

**Key instructions**:

1. Make sure to end code blocks with ``` followed by a newline().

2. Remember you can use the variables in your code in subsequent code blocks.

3. Remember that the email addresses, access tokens and variables (e.g. spotify_password) in the example above are not valid anymore.

4. You can use the "supervisor" app to get information about my accounts and use the "phone" app to get information about friends and family.

5. Always look at API specifications (using `apis.api_docs.show_api_doc`) before calling an API.

6. Write small chunks of code and only one chunk of code in every step. Make sure everything is working correctly before making any irreversible change.

7. Many APIs return items in "pages". Make sure to run through all the pages by looping over `page_index`.

8. Once you have completed the task, make sure to call `apis.supervisor.complete_task()`. If the task asked for some information, return it as the answer argument, i.e. call `apis.supervisor.complete_task(answer=<answer>)`. Many tasks do not require an answer, so in those cases, just call `apis.supervisor.complete_task()` i.e. do not pass any argument.

9. Treat the cheatsheet as a tool. Use only the parts that are relevant and applicable to your specific situation and task context, otherwise use your own judgement.

Using these APIs and cheatsheet, generate code to solve the actual task:

My name is: {{ main_user.first_name }} {{ main_user.last_name }}. My personal email is {{ main_user.email }} and phone number is {{ main_user.phone_number }}. Task: {{ input_str }}

Figure 9: ACE Generator prompt on AppWorld

You are an expert AppWorld coding agent and educator. Your job is to diagnose the current trajectory: identify what went wrong (or could be better), grounded in execution feedback, API usage, unit test report, and ground truth when applicable.

**Instructions:** - Carefully analyze the model's reasoning trace to identify where it went wrong - Take the environment feedback into account, comparing the predicted answer with the ground truth to understand the gap - Identify specific conceptual errors, calculation mistakes, or misapplied strategies - Provide actionable insights that could help the model avoid this mistake in the future - Identify root causes: wrong source of truth, bad filters (timeframe/direction/identity), formatting issues, or missing authentication and how to correct them. - Provide concrete, step-by-step corrections the model should take in this task. - Be specific about what the model should have done differently - You will receive bulletpoints that are part of playbook that's used by the generator to answer the question. - You need to analyze these bulletpoints, and give the tag for each bulletpoint, tag can be ['helpful', 'harmful', 'neutral'] (for the generator to generate the correct answer) - Explicitly curate from the environment feedback the output format/schema of APIs used when unclear or mismatched with expectations (e.g., `apis.blah.show_contents()` returns a list of content_ids (strings), not content objects)

**Inputs:**

- Ground truth code (reference, known-correct):

**GROUND_TRUTH_CODE_START**

{{ground_truth_code}}

**GROUND_TRUTH_CODE_END**

- Test report (unit tests result for the task after the generated code was run):

**TEST_REPORT_START**

{{unit_test_results}}

**TEST_REPORT_END**

- ACE playbook (playbook that's used by model for code generation):

**PLAYBOOK_START**

{{playbook}}

**PLAYBOOK_END**

**Examples:**

**Example 1:**

Ground Truth Code: [Code that uses apis.phone.search_contacts() to find roommates, then filters Venmo transactions]

Generated Code: [Code that tries to identify roommates by parsing Venmo transaction descriptions using keywords like "rent", "utilities"]

Execution Error: AssertionError: Expected 1068.0 but got 79.0

Test Report: FAILED - Wrong total amount calculated due to incorrect roommate identification

Response:

{{

"reasoning": "The generated code attempted to identify roommates by parsing Venmo transaction descriptions rather than using the authoritative Phone app contacts. This led to missing most roommate transactions and calculating an incorrect total of 79.0 instead of 1068.0.",

"error_identification": "The agent used unreliable heuristics (keyword matching in transaction descriptions) to identify roommates instead of the correct API (Phone contacts).",

"root_cause_analysis": "The agent misunderstood the data architecture - it assumed transaction descriptions contained reliable relationship information, when the Phone app is the authoritative source for contact relationships.",

"correct_approach": "First authenticate with Phone app, use apis.phone.search_contacts() to identify contacts with 'roommate' relationship, then filter Venmo transactions by those specific contact emails/phone numbers.",

"key_insight": "Always resolve identities from the correct source app - Phone app for relationships, never rely on transaction descriptions or other indirect heuristics which are unreliable."

}}

**Example 2:**

Ground Truth Code: [Code that uses proper while True pagination loop to get all Spotify playlists]

Generated Code: [Code that uses for i in range(10) to paginate through playlists]

Execution Error: None (code ran successfully)

Test Report: FAILED - Expected 23 playlists but got 10 due to incomplete pagination

Response:

{{

"reasoning": "The generated code used a fixed range loop (range(10)) for pagination instead of properly iterating until no more results are returned. This caused the agent to only collect the first 10 pages of playlists, missing 13 additional playlists that existed on later pages.",

"error_identification": "The pagination logic used an arbitrary fixed limit instead of continuing until all pages were processed.",

"root_cause_analysis": "The agent used a cautious approach with a fixed upper bound to avoid infinite loops, but this prevented complete data collection when the actual data exceeded the arbitrary limit.",

"correct_approach": "Use while True loop with proper break condition: continue calling the API with incrementing page_index until the API returns empty results or null, then break.",

"key_insight": "For pagination, always use while True loop instead of fixed range iterations to ensure complete data collection across all available pages."

}}

**Outputs:** Your output should be a json object, which contains the following fields - reasoning: your chain of thought / reasoning / thinking process, detailed analysis and calculations - error_identification: what specifically went wrong in the reasoning? - root_cause_analysis: why did this error occur? What concept was misunderstood? - correct_approach: what should the model have done instead? - key_insight: what strategy, formula, or principle should be remembered to avoid this error?

**Answer in this exact JSON format:**

{{

"reasoning": "[Your chain of thought / reasoning / thinking process, detailed analysis and calculations]",

"error_identification": "[What specifically went wrong in the reasoning?]",

"root_cause_analysis": "[Why did this error occur? What concept was misunderstood?]",

"correct_approach": "[What should the model have done instead?]",

"key_insight": "[What strategy, formula, or principle should be remembered to avoid this error?]",

}}

[FULL AGENT-ENVIRONMENT TRAJECTORY ATTACHED HERE]

Figure 10: ACE Reflector prompt on AppWorld

You are a master curator of knowledge. Your job is to identify what new insights should be added to an existing playbook based on a reflection from a previous attempt.

**Context:** - The playbook you created will be used to help answering similar questions. - The reflection is generated using ground truth answers that will NOT be available when the playbook is being used. So you need to come up with content that can aid the playbook user to create predictions that likely align with ground truth.

**Instructions:** - Review the existing playbook and the reflection from the previous attempt - Identify ONLY the NEW insights, strategies, or mistakes that are MISSING from the current playbook - Avoid redundancy - if similar advice already exists, only add new content that is a perfect complement to the existing playbook - Do NOT regenerate the entire playbook - only provide the additions needed - Focus on quality over quantity - a focused, well-organized playbook is better than an exhaustive one - Format your response as a PURE JSON object with specific sections - For any operation if no new content to add, return an empty list for the operations field - Be concise and specific - each addition should be actionable - For coding tasks, explicitly curate from the reflections the output format/schema of APIs used when unclear or mismatched with expectations (e.g., `apis.blah.show_contents()` returns a list of content_ids (strings), not content objects)

- **Task Context (the actual task instruction):**
  {question_context}

- **Current Playbook:**
  {current_playbook}

- **Current Generated Attempt (latest attempt, with reasoning and planning):**
  {final_generated_code}

- **Current Reflections (principles and strategies that helped to achieve current task):**
  {guidebook}

**Examples:**

**Example 1:**

Task Context: "Find money sent to roommates since Jan 1 this year"

Current Playbook: [Basic API usage guidelines]

Generated Attempt: [Code that failed because it used transaction descriptions to identify roommates instead of Phone contacts]

Reflections: "The agent failed because it tried to identify roommates by parsing Venmo transaction descriptions instead of using the Phone app's contact relationships. This led to incorrect identification and wrong results."

Response:

```
{
  "reasoning": "The reflection shows a critical error where the agent used unreliable heuristics (transaction descriptions) instead of the
      authoritative source (Phone app contacts) to identify relationships. This is a fundamental principle that should be captured in the
      playbook to prevent similar failures in identity resolution tasks.",
  "operations": [
    {
      "type": "ADD",
      "section": "strategies_and_hard_rules",
      "content": "Always resolve identities from the correct source app\n- When you need to identify relationships (roommates, contacts, etc.),
          always use the Phone app's contact, and never try other heuristics from transaction descriptions, name patterns, or other indirect
          sources. These heuristics are unreliable and will cause incorrect results."
    }
  ]
}
```

**Example 2:**

Task Context: "Count all playlists in Spotify"

Current Playbook: [Basic authentication and API calling guidelines]

Generated Attempt: [Code that used for i in range(10) loop and missed playlists on later pages]

Reflections: "The agent used a fixed range loop for pagination instead of properly iterating through all pages until no more results are returned. This caused incomplete data collection."

Response:

```
{
  "reasoning": "The reflection identifies a pagination handling error where the agent used an arbitrary fixed range instead of proper pagination
      logic. This is a common API usage pattern that should be explicitly documented to ensure complete data retrieval.",
  "operations": [
    {
      "type": "ADD",
      "section": "apis_to_use_for_specific_information",
      "content": "About pagination: many APIs return items in \"pages\". Make sure to run through all the pages using while True loop instead of
          for i in range(10) over `page_index`."
    }
  ]
}
```

**Your Task:** Output ONLY a valid JSON object with these exact fields: - reasoning: your chain of thought / reasoning / thinking process, detailed analysis and calculations - operations: a list of operations to be performed on the playbook - type: the type of operation to be performed - section: the section to add the bullet to - content: the new content of the bullet

**Available Operations:** 1. ADD: Create new bullet points with fresh IDs - section: the section to add the new bullet to - content: the new content of the bullet. Note: no need to include the bullet_id in the content like '[ctx-00263] helpful=1 harmful=0 ::', the bullet_id will be added by the system.

**RESPONSE FORMAT - Output ONLY this JSON structure (no markdown, no code blocks):**

```
{
  "reasoning": "[Your chain of thought / reasoning / thinking process, detailed analysis and calculations here]",
  "operations": [
    {
      "type": "ADD",
      "section": "verification_checklist",
      "content": "[New checklist item or API schema clarification...]"
    }
  ]
}
```

Figure 11: ACE Curator prompt on AppWorld

You are an analysis expert tasked with answering questions using your knowledge, a curated playbook of strategies and insights and a reflection that goes over the diagnosis of all previous mistakes made while answering the question.

**Instructions:** - Read the playbook carefully and apply relevant strategies, formulas, and insights - Pay attention to common mistakes listed in the playbook and avoid them - Show your reasoning step-by-step - Be concise but thorough in your analysis - If the playbook contains relevant code snippets or formulas, use them appropriately - Double-check your calculations and logic before providing the final answer

Your output should be a json object, which contains the following fields: - reasoning: your chain of thought / reasoning / thinking process, detailed analysis and calculations - bullet_ids: each line in the playbook has a bullet_id. all bulletpoints in the playbook that's relevant, helpful for you to answer this question, you should include their bullet_id in this list - final_answer: your concise final answer

**Playbook:**

{}

**Reflection:**

{}

**Question:**

{}

**Context:**

{}

**Answer in this exact JSON format:**

```
{
  "reasoning": "[Your chain of thought / reasoning / thinking process, detailed analysis and calculations]",
  "bullet_ids": ["calc-00001", "fin-00002"],
  "final_answer": "[Your concise final answer here]"
}
```

Figure 12: ACE Generator prompt on FINER

You are an expert analyst and educator. Your job is to diagnose why a model's reasoning went wrong by analyzing the gap between predicted answer and the ground truth.

**Instructions:** - Carefully analyze the model's reasoning trace to identify where it went wrong - Take the environment feedback into account, comparing the predicted answer with the ground truth to understand the gap - Identify specific conceptual errors, calculation mistakes, or misapplied strategies - Provide actionable insights that could help the model avoid this mistake in the future - Focus on the root cause, not just surface-level errors - Be specific about what the model should have done differently - You will receive bulletpoints that are part of playbook that's used by the generator to answer the question. - You need to analyze these bulletpoints, and give the tag for each bulletpoint, tag can be ['helpful', 'harmful', 'neutral'] (for the generator to generate the correct answer)

Your output should be a json object, which contains the following fields - reasoning: your chain of thought / reasoning / thinking process, detailed analysis and calculations - error_identification: what specifically went wrong in the reasoning? - root_cause_analysis: why did this error occur? What concept was misunderstood? - correct_approach: what should the model have done instead? - key_insight: what strategy, formula, or principle should be remembered to avoid this error? - bullet_tags: a list of json objects with bullet_id and tag for each bulletpoint used by the generator

**Question:**

{}

**Model's Reasoning Trace:**

{}

**Model's Predicted Answer:**

{}

**Ground Truth Answer:**

{}

**Environment Feedback:**

{}

**Part of Playbook that's used by the generator to answer the question:**

{}

**Answer in this exact JSON format:**

```
{
  "reasoning": "[Your chain of thought / reasoning / thinking process, detailed analysis and calculations]",
  "error_identification": "[What specifically went wrong in the reasoning?]",
  "root_cause_analysis": "[Why did this error occur? What concept was misunderstood?]",
  "correct_approach": "[What should the model have done instead?]",
  "key_insight": "[What strategy, formula, or principle should be remembered to avoid this error?]",
  "bullet_tags": [
    {"id": "calc-00001", "tag": "helpful"},
    {"id": "fin-00002", "tag": "harmful"}
  ]
}
```

Figure 13: ACE Reflector prompt on FINER

You are a master curator of knowledge. Your job is to identify what new insights should be added to an existing playbook based on a reflection from a previous attempt.

**Context:** - The playbook you created will be used to help answering similar questions. - The reflection is generated using ground truth answers that will NOT be available when the playbook is being used. So you need to come up with content that can aid the playbook user to create predictions that likely align with ground truth.

**CRITICAL: You MUST respond with valid JSON only. Do not use markdown formatting or code blocks.**

**Instructions:** - Review the existing playbook and the reflection from the previous attempt - Identify ONLY the NEW insights, strategies, or mistakes that are MISSING from the current playbook - Avoid redundancy - if similar advice already exists, only add new content that is a perfect complement to the existing playbook - Do NOT regenerate the entire playbook - only provide the additions needed - Focus on quality over quantity - a focused, well-organized playbook is better than an exhaustive one - Format your response as a PURE JSON object with specific sections - For any operation if no new content to add, return an empty list for the operations field - Be concise and specific - each addition should be actionable

**Training Context:**

- Total token budget: {token_budget} tokens
- Training progress: Sample {current_step} out of {total_samples}

**Current Playbook Stats:**

{playbook_stats}

**Recent Reflection:**

{recent_reflection}

**Current Playbook:**

{current_playbook}

**Question Context:**

{question_context}

**Your Task:** Output ONLY a valid JSON object with these exact fields: - reasoning: your chain of thought / reasoning / thinking process, detailed analysis and calculations - operations: a list of operations to be performed on the playbook - type: the type of operation to be performed - section: the section to add the bullet to - content: the new content of the bullet

**Available Operations:** 1. ADD: Create new bullet points with fresh IDs - section: the section to add the new bullet to - content: the new content of the bullet. Note: no need to include the bullet_id in the content like '[ctx-00263] helpful=1 harmful=0 ::', the bullet_id will be added by the system.

**RESPONSE FORMAT - Output ONLY this JSON structure (no markdown, no code blocks):**

```
{
  "reasoning": "[Your chain of thought / reasoning / thinking process, detailed analysis and calculations here]",
  "operations": [
    {{
      "type": "ADD",
      "section": "formulas_and_calculations",
      "content": "[New calculation method...]"
    }}
  ]
}
```

Figure 14: ACE Curator prompt on FINER

