# OpenReview forum: "Agentic Context Engineering: Evolving Contexts for Self-Improving Language Models"
_ICLR.cc/2026/Conference — ICLR 2026 Poster_

### Official Review · Reviewer_LJpr · 2025-10-22

**Soundness:** 2
**Presentation:** 2
**Contribution:** 2
**Rating:** 4
**Confidence:** 3

**Summary:**

This paper introduces ACE, a framework where the model adjusts its inputs using “reference materials” such as task instructions, coping strategies, or past experiences. By leveraging these contextual references, ACE helps improve the model’s overall performance.

**Strengths:**

1. I really like Section 2.2 — it clearly sets up the motivation by analyzing Brevity Bias and Context Collapse before presenting the problem.

2. The proposed method looks quite effective.

**Weaknesses:**

1. The method heavily depends on feedback quality, but lacks a clear solution for unreliable feedback and has limited robustness evaluation.

2. The causal attribution of each core module’s effectiveness is unclear, and the ablation study is somewhat incomplete.

3. The discussion on long-context cost is not convincing enough and doesn’t fully reflect real-world deployment concerns.

**Questions:**

1. The paper states that ACE relies heavily on reliable feedback signals (e.g., code execution results or ground-truth labels), but doesn’t explain what happens when feedback quality is poor. Experiments show that when there are no gold labels or the feedback is noisy (like in ambiguous financial tasks), ACE’s performance drops sharply—sometimes even below baseline. However, this is only listed as a “limitation,” with no mitigation strategies explored. The paper also lacks quantitative tests on low-quality feedback (e.g., different noise levels), so we can’t tell how robust ACE is in messy real-world environments where feedback can be unreliable.

2. The current ablation only compares “with vs. without reflection” and “single-round vs. multi-round adaptation,” but doesn’t isolate the two key designs — incremental update and refine-as-you-extend. As a result, it’s unclear whether the gains come mainly from reducing information loss (via incremental updates), cutting redundancy (via refine-as-you-extend), or both. This makes it harder for future research to pinpoint which component drives the improvement.

3. The paper doesn’t test what happens when the generator, reflector, and summarizer use models of different strengths. All components use the same model (DeepSeek-V3.1), but in real scenarios, one might use a stronger model for the reflector. Would ACE still be stable in that case? Would costs rise significantly? These are important but unexplored questions.

4. The paper mentions using “KV cache reuse” to reduce cost, but doesn’t actually show numbers. For instance, how does ACE’s inference time or memory usage change when handling 18k-token contexts with and without caching? How does that compare to short-context methods (like GEPA’s concise prompts)? Without this data, the claim of “low cost” feels unsubstantiated for practical readers.

5. The paper also skips the retrieval efficiency problem for long contexts. ACE’s context is a structured “playbook,” but as it grows, how does the model quickly find the relevant parts (e.g., specific financial rules)? If retrieval time increases, it could cancel out the latency benefits of incremental updates. The paper doesn’t discuss or test any such optimization.

6. In the AppWorld experiments, ACE is compared against a “top commercial agent (IBM-CUGA)” that’s said to be “based on GPT-4.1.” However, the paper doesn’t reveal whether that system also used context optimization or even had the same test setup. If not, ACE’s “catch-up” advantage might be overstated, raising concerns about fairness.

7. In ACE’s refine-as-you-extend mechanism, key parameters like the deduplication threshold (how to judge when two strategy fragments are semantically redundant), maximum context length (when to trigger pruning), and number of reflection iterations (set to 5) are not specified or justified. Without details on why “5 rounds” were chosen or how different values affect performance, it’s hard for others to reproduce the reported results accurately.

---

> ### Author Response · Authors · 2025-11-21
> **Rebuttal to reviewer LJpr [1]**
>
> We thank the reviewer for the kind words and helpful feedback! We address each concern/question below.
>
> **#1. Sensitivity to reflection quality.**
>
> We thank the reviewer for raising this point. We performed two additional analyses to directly evaluate ACE’s sensitivity to reflection quality. Both experiments use offline adaptation with ACE.
>
> (1) **Using Weaker Reflector Models.** To test whether ACE requires a strong reflector, we vary the reflector LLM across three models with substantial differences in size and capability: GPT-OSS-120B, DeepSeek-V3.1-671B, and GPT-5.1. Across all settings, ACE consistently improves over the base LLM, even when the reflector is considerably weaker. Stronger reflectors do yield larger gains, but ACE remains beneficial with a wide range of model choices.
>
> | Method | Generator LLM | Reflector LLM | Curator LLM | Accuracy |
> |---------|---------|---------|---------|---------|
> | Base LLM | DeepSeek-V3.1 | - | - | 70.7 |
> | ACE | DeepSeek-V3.1 | GPT-OSS-120b  | DeepSeek-V3.1 | 76.6 (+5.9) |
> | ACE | DeepSeek-V3.1 | DeepSeek-V3.1  | DeepSeek-V3.1 | 78.3 (+7.6) |
> | ACE | DeepSeek-V3.1 | GPT-5.1 | DeepSeek-V3.1 | 78.5 (+7.8) |
>
> (2) **Robustness to Noisy or Harmful Reflector Feedback.** Next, we evaluate ACE under actively harmful or noisy reflector outputs. We inject adversarial or conflicting bullets into the playbook by invoking a “harmful” reflector once every X adaptation steps. A larger X means less frequent noise. We observe that **ACE is robust to moderate noise levels**: performance degrades only gradually as noise increases and remains above the base LLM unless the harmful reflector is invoked every iteration. This provides direct evidence that ACE tolerates substantial noise in the reflection stream and uses a stable update mechanism; only extreme, adversarial corruption causes noticeable degradation.
>
> | Harmful Reflector Frequency (every X iters) | Accuracy |
> |---------|---------|
> | base LLM | 70.7 |
> | 1 | 66.7 (-4.0)  |
> | 5 | 76.1 (+5.4) |
> | 10 | 77.0 (+6.3) |
> | 25 | 77.8 (+7.1) |
> | 50 | 78.2 (+7.5) |
> | 100 | 78.2 (+7.5) |
> | No harmful reflector | 78.3 (+7.6) |
>
> Overall, ACE is not highly sensitive to reflection quality: (1) Even weaker LLMs produce substantial gains. (2) ACE remains stable under moderate noise or conflicting updates. (3) Performance only drops below the base model under intentionally adversarial conditions.
>
> **Noise mitigation strategies:** ACE’s bullet point analyzer (also named “grow-and-refine” in Section 3.2), which performs merging, deduplication, and conflict resolution, acts as the first line of defense against context noise. When triggered (for example, after each curator call or when the playbook exceeds a size threshold), the analyzer examines pairs of semantically similar bullet points, removes duplicates, and filters out entries marked as potentially harmful (that is, those with high “harmful” counters in the metadata). Additional noise-mitigation strategies, such as detecting contradictory bullets, encouraging the curator to prioritize high-confidence updates, or periodically pruning outdated entries, could further keep the playbook compact, consistent, and reliable. These noise-mitigation strategies are compatible additions to the ACE framework. We will add more discussion of these mechanisms in the final version.
>
> **#2. Additional ablation studies of ACE.**
>
> We thank the reviewer for suggesting further ablations to better isolate the sources of ACE’s performance gains. We performed an additional study on incremental context update (results below); we will include additional results on grow-and-refine ablation in the final version of the paper.
>
> In this ablation, we run offline context adaptation on AppWorld using ACE with and without incremental updates, and report evaluation results on the test-normal split with DeepSeek-V3.1:
>
> | Method                       | test-normal TGC           | test-normal SGC           | Average            |
> |------------------------------|----------------------------|----------------------------|----------------------|
> | Base ReAct                   | 61.9                       | 46.4                       | 54.2                 |
> | ACE (no incremental update)  | 67.3 (+5.4)                | 46.4 (+0.0)                | 56.9 (+2.7)          |
> | ACE (with incremental update)| 76.2 (+14.3)               | 64.3 (+17.9)               | 70.3 (+16.1)         |
>
> The initial findings indicate that incremental context update, by preserving useful information that would otherwise be lost due to context collapse, is a central contributor to ACE’s improvement.

---

> ### Author Response · Authors · 2025-11-21
> **Rebuttal to reviewer LJpr [2]**
>
> **#3. The use of heterogeneous LLMs within ACE.**
>
> We thank the reviewer for highlighting the practical scenario in which the Reflector may be a stronger model than the Generator and the Curator. To evaluate this, we conducted additional experiments using a broader set of Reflector LLMs spanning different sizes and capability levels (GPT-OSS-120B, DeepSeek-V3.1, GPT-5.1). The results below show that ACE consistently delivers substantial gains across all configurations. While stronger Reflector models generally yield larger improvements, ACE remains effective even with more modest Reflectors, demonstrating robustness to heterogeneous LLM choices.
>
> | Method | Generator LLM | Reflector LLM | Curator LLM | Accuracy |
> |---------|---------|---------|---------|---------|
> | Base LLM | DeepSeek-V3.1 | - | - | 70.7 |
> | ACE | DeepSeek-V3.1 | GPT-OSS-120b  | DeepSeek-V3.1 | 76.6 (+5.9) |
> | ACE | DeepSeek-V3.1 | DeepSeek-V3.1  | DeepSeek-V3.1 | 78.3 (+7.6) |
> | ACE | DeepSeek-V3.1 | GPT-5.1 | DeepSeek-V3.1 | 78.5 (+7.8) |
>
> These results indicate that ACE is flexible in practice: it benefits from strong Reflectors when available, yet continues to provide reliable improvements even in fully homogeneous or resource-constrained settings.
>
> **#4. KV cache reuse and inference cost.**
>
> We thank the reviewer for raising this point. We conducted a more fine-grained cost analysis to examine how prompt caching affects ACE’s inference cost, focusing on the evaluation stage.
>
> After generating the ACE playbook via offline adaptation, we evaluated ACE, GEPA, and baseline methods on AppWorld’s test-normal split using GPT-5.1 with OpenAI’s default prompt-caching mechanism. We find that **91.8% of ACE’s 63.7M input tokens (58.5M tokens)** were served from cache and billed at only 10% of standard price, yielding an **82.6% reduction in input-token dollar cost**. This supports our claim that longer contexts do not necessarily incur dramatically higher serving cost when prompt caching is available.
>
> **Additional notes:** (1) Adaptation-stage caching. ACE also benefits from caching during adaptation. Each delta update is inserted at a semantically appropriate location within the playbook. The prefix before the insertion point can fully reuse the KV cache, while the region starting at the modified location must be recomputed. In practice, delta updates modify only a small, localized portion of the playbook, so the reusable prefix is large, yielding substantial prefill savings. Bullet-point merging and deduplication represent a special case: these operations may rewrite a larger span of text, reducing the reusable prefix for that step. However, such events are infrequent, and even then, the prefix up to the section boundary remains valid and can be cached.
> (2) Recent systems work has also explored KV-cache reuse even under changed positional encodings, which could reduce the inference cost of ACE even more. We view such advances, along with inference acceleration and partial-context sharing, as promising directions for further optimizing ACE.
>
> **#5. Retrieval efficiency.**
>
> We thank the reviewer for raising this concern. ACE, in its current form, does not rely on an explicit retrieval module (e.g. embedding-based retrieval or LM-as-retriever). Instead, the “retrieval cost” corresponds to the prefill cost of loading the structured playbook into the KV cache before decoding begins. As discussed in #4, prompt caching substantially reduces this cost: the majority of the playbook tokens are served from cache, and only a small delta must be recomputed when updates occur, yielding large prefill and dollar-cost savings in practice.
>
> While the vanilla ACE framework performs well without a dedicated retriever, it can naturally incorporate one. For instance, the playbook can be selectively truncated or filtered using embedding similarity or section-level relevance scoring, feeding only the most relevant subset to the model. We view these retrieval-optimization mechanisms, together with partial-context sharing and KV-cache techniques, as promising extensions to the ACE framework.

---

> ### Author Response · Authors · 2025-11-21
> **Rebuttal to reviewer LJpr [3]**
>
> **#6. Clarification on comparison with IBM-CUGA.**
>
> Our mention of IBM-CUGA is intended only as a rough contextual benchmark, illustrating that ACE reaches the performance range of a widely used agent on the AppWorld leaderboard. It is not used as a methodological baseline, nor do we draw any conclusions from direct, apples-to-apples comparison. The internal design choices of CUGA differ from the context-adaptation focus of ACE, and the two systems target somewhat different aspects of agent behavior.
>
> All baselines included in our experiments (e.g., Base ReAct, GEPA) are **methodological baselines**, they isolate different approaches to prompt or context construction, and are evaluated under identical setups. This allows us to attribute performance differences directly to methodological choices rather than differences in agent engineering.
>
> We will revise the paper to clarify these points. Because ACE is method-level and modular, integrating it with systems such as IBM-CUGA is a natural direction for future work.
>
> **#7. Sensitivity to hyper-parameters in multi-round reflection and grow-and-refine.**
>
> We provide additional sensitivity studies for ACE’s key hyper-parameters, including the number of reflection iterations, the deduplication threshold, and the maximum context length for triggering pruning.
>
> For the number of reflection iterations, our goal is to balance two effects: (1) extracting enough high-quality insights from the Reflector’s traces, and (2) avoiding overthinking that may introduce unnecessary or noisy updates. Empirically, “5 rounds” provides a strong balance. As shown below (AppWorld results), 1 round under-extracts useful strategy fragments and leaves substantial room for improvement, while too many rounds (e.g. 10) could lead to overthinking and degrade performance.
>
> | Number of reflection iterations | test-normal TGC | test-normal SGC | Average |
> |----------------------------|------------------|------------------|---------|
> | 1  | 69.0 | 53.6 | 61.3 |
> | 3  | 74.4 | 57.1 | 65.8 |
> | 5  | 72.6 | 62.5 | 67.6 |
> | 10 | 71.4 | 58.9 | 65.2 |
>
> For the deduplication threshold and the maximum context length for triggering pruning, our goal is to regulate the growth of the evolving context without removing genuinely useful insights. These two hyper-parameters control complementary aspects of ACE’s grow-and-refine loop:
> - Deduplication threshold: This governs how aggressively ACE merges newly extracted insights with existing ones. The FiNER results show that performance varies only mildly across this range, indicating that ACE’s reflection-curation pipeline is robust to moderate changes in dedup aggressiveness.
>
> | Deduplication threshold | FiNER Accuracy |
> |----------------------------|------------------|
> | 50% | 77.0 |
> | 70% | 73.9 |
> | 90% | 78.6 |
>
> - Maximum context length for pruning: This parameter defines when ACE triggers merging and pruning of older or less useful entries to prevent unbounded growth. The trade-off is again between (1) allowing enough accumulated context to improve "learning from experience", and (2) keeping the context compact enough to avoid unnecessary prompt-processing cost or noise accumulation. Across a wide range, from 10K to 100K tokens, FiNER performance remains stable, showing that ACE does not rely on finely tuned length thresholds. This is because pruning primarily removes stale or low-utility fragments, while the reflective updates preserve the core reusable strategies.
>
> | Maximum context length | FiNER Accuracy |
> |----------------------------|------------------|
> | 10K | 78.6 |
> | 50K | 78.4 |
> | 100K | 78.3 |
>
> Together, these results show that ACE is not overly sensitive to the exact values of its key hyper-parameters. Reasonable settings (e.g. 3–5 reflection rounds, 50–90% dedup threshold, and 10K–100K pruning triggers) all lead to strong and stable performance. We will include these analyses in the final version to support full reproducibility.
>
> **Finally:** We really appreciate your insightful suggestions, which have already helped us strengthen both the analysis and presentation of this work. We hope the above clarifications resolve the concerns. If the response is helpful, we would be grateful if you could consider adjusting your score accordingly. We are committed to incorporating all promised additions in the camera‑ready version to make the paper as valuable as possible to the community.

---

> > ### Comment · Reviewer_LJpr · 2025-11-25
> >
> > Thank you for the detailed and well-organized rebuttal. I am genuinely impressed by the clarity and completeness of your response. The additional experiments and analyses directly address the core weaknesses I raised in my initial review, and they substantially improve my confidence in both the methodology and the conclusions of the paper. Given the strength of the rebuttal and the substantial improvement in both clarity and empirical grounding, I am happy to revise my score from 4 to 8.
> > The work is now convincing and, in my view, clearly merits acceptance.
> > I appreciate the authors’ careful and constructive response, and I believe the paper will make a valuable contribution once the promised revisions are incorporated.

---

> > > ### Author Response · Authors · 2025-11-28
> > > **Thank you!**
> > >
> > > Dear Reviewer,
> > >
> > > Thank you so much for your reply!
> > >
> > > We really appreciate your feedback and suggestions, and we are committed to incorporating these results and discussions into the final version of the paper.
> > >
> > > Sincerely,
> > > Authors

---

### Official Review · Reviewer_KcGQ · 2025-10-26

**Soundness:** 2
**Presentation:** 3
**Contribution:** 2
**Rating:** 4
**Confidence:** 4

**Summary:**

The paper introduces ACE (Agentic Context Engineering), a framework for context-based self-improvement of LLMs that treats prompts as evolving, structured playbooks rather than concise summaries. It addresses two common failures in prior prompt optimization—brevity bias and context collapse—by separating roles into a Generator, Reflector, and Curator; applying incremental delta updates instead of monolithic rewrites; and using a grow-and-refine mechanism for de-duplication and maintenance. ACE operates in both offline (prompt/system prompt optimization) and online (test-time memory) settings, leveraging natural execution feedback and optionally ground-truth labels. Across agent (AppWorld) and domain-specific (FiNER, Formula) benchmarks, ACE outperforms strong baselines (e.g., GEPA, DC), delivering average gains around 9–11%, matching or exceeding a top production agent on AppWorld using a smaller open-source model, and substantially reducing adaptation latency and cost (e.g., up to ~82–92% latency reduction). Ablations confirm the value of the Reflector, multi-epoch refinement, and offline warmup. Limitations include reliance on feedback quality and a capable Reflector, and reduced benefits in tasks that favor concise instructions over rich, accumulated context.

**Strengths:**

1. The paper is in general well-written.
2. The benchmarks are carefully selected and representative, but would be better to include more to make the conclusions more solid, e.g., 1 or 2 more.
3. The paper identifies two important pitfalls in existing works, and designs better approaches to address the issue.

**Weaknesses:**

1. It is not clear that whether the proposed will still be effective for more powerful (and potentially more knowledgeable model like GPT-5)
2. It is not accurate to claim that ACE uses much smaller model (DeepSeek-V3.1, 685B) than GPT-4.1 (unknown size)
3. This work builds on top of Dynamic cheatsheet. More detailed comparison and analysis to it is necessary. What is the main difference to it? What are their scores in the benchmark? Currently, I feel that ACE makes minore changes, while the core idea to evolving playbooks s that accumulate, refine, and organize strategies is similar to Dynamic cheatsheet.

**Questions:**

Could you provide some examples on context collapse as shown in Figure 2?

---

> ### Author Response · Authors · 2025-11-21
> **Rebuttal to reviewer KcGQ [1]**
>
> We thank the reviewer for the kind words and helpful feedback! We address each concern/question below.
>
> **#1. Generalization to language models beyond DeepSeek-V3.1.**
>
> ACE is a model-agnostic framework that operates on execution traces and contextual deltas, and does not rely on any architectural or training-specific features of DeepSeek-V3.1. We evaluated ACE with three additional models of varying size, cost, and capability: GPT-OSS-120B, GPT-5.1, and Llama-3.3-70B-Instruct. In each case, the Generator, Reflector, and Curator were all switched to the new model without any changes to the algorithm.
>
> **(1) GPT-OSS-120b**
>
> AppWorld
> | Method                                   | test-normal TGC | test-normal SGC | test-challenge TGC | test-challenge SGC | Average     |
> |------------------------------------------|------------------|------------------|---------------------|---------------------|-------------|
> | Base ReAct                               | 54.8             | 33.9             | 34.5                | 15.1                | 34.6        |
> | GEPA                                     | 56.0 (+1.2)      | 33.9 (+0.0)      | 40.1 (+5.6)         | 20.9 (+5.8)         | 37.7 (+3.1) |
> | ACE (offline, with ground-truth labels)  | 61.3 (+6.5)      | 39.3 (+5.4)      | 40.3 (+5.8)         | 20.9 (+5.8)         | 40.5 (+5.9) |
> | ACE (offline, no ground-truth labels)    | 58.3 (+3.5)      | 41.1 (+7.2)      | 39.6 (+5.1)         | 18.7 (+3.6)         | 39.4 (+4.8) |
> | DC                                       | 49.4 (-5.4)      | 33.9 (+0.0)      | 30.8 (-3.7)                   | 18.2 (+3.1)                   | 33.1 (-1.5)           |
> | ACE (online)                             | 60.7 (+5.9)      | 44.6 (+10.7)     | 43.2 (+8.7)         | 20.1 (+5.0)         | 42.2 (+7.6) |
>
> Finance
> | Method                                      | FINER Accuracy     | Formula Accuracy    | Average       |
> |---------------------------------------------|---------------------|----------------------|---------------|
> | Base LLM                                    | 66.6                | 71.5                 | 69.1          |
> | GEPA                                        | 67.9 (+1.3)         | 71.5 (+0.0)          | 69.7 (+0.6)   |
> | ACE (offline, with ground-truth labels)     | 73.8 (+7.2)         | 88.5 (+17.0)         | 81.2 (+12.1)  |
> | ACE (offline, no ground-truth labels)       | 69.7 (+3.1)         | 84.0 (+12.5)         | 76.9 (+7.8)   |
> | DC                                          | 55.8 (-10.8)        | 66.0 (-5.5)          | 60.9 (-8.2)   |
> | ACE (online)                                | 70.5 (+3.9)         | 85.0 (+13.5)         | 77.8 (+8.7)   |
>
> **(2) GPT-5.1**
>
> AppWorld
> | Method                     | test-normal TGC          | test-normal SGC          | Average             |
> |----------------------------|---------------------------|----------------------------|----------------------|
> | Base ReAct                 | 61.9                      | 46.4                       | 54.2                |
> | GEPA                       | 64.3 (+2.4)               | 48.2 (+1.8)                | 56.2 (+2.0)        |
> | ACE (offline, with ground-truth labels)       | 66.7 (+4.8)               | 53.6 (+7.2)                | 60.2 (+6.0)        |
> | ACE (offline, no ground-truth labels)      | 67.3 (+5.4)               | 55.4 (+9.0)                | 61.3 (+7.1)        |
> | DC                         | 62.5 (+0.6)               | 55.4 (+9.0)                | 58.9 (+4.7)        |
> | ACE (online)               | 72.6 (+10.7)              | 58.9 (+12.5)               | 65.8 (+11.6)       |
>
> Finance
> | Method                     | FINER Accuracy        | Formula Accuracy       | Average             |
> |----------------------------|------------------------|-------------------------|----------------------|
> | Base LLM                   | 73.5                  | 73.0                   | 73.3                |
> | GEPA                       | 74.5 (+1.0)           | 73.0 (+0.0)            | 73.8 (+0.5)         |
> | ACE (offline, with ground-truth labels)       | 81.0 (+7.5)           | 84.5 (+11.5)           | 82.8 (+9.5)         |
> | ACE (offline, no ground-truth labels)      | 78.2 (+4.7)           | 76.5 (+3.5)            | 77.4 (+4.1)         |
> | DC                         |   70.0 (-3.5)                  | 69.0 (-4.0)                      |  69.5 (-3.8)                 |
> | ACE (online)               | 75.2 (+1.7)           | 79.0 (+6.0)            | 77.1 (+3.8)         |

---

> ### Author Response · Authors · 2025-11-21
> **Rebuttal to reviewer KcGQ [2]**
>
> **(3) Llama-3.3-70b-Instruct**
>
> Finance
> | Method                     | FINER Accuracy        |
> |----------------------------|------------------------|
> | Base LLM                      | 62.5                   |
> | GEPA                       | 59.41 (-3.09)          |
> | ACE (offline, with ground-truth labels)       | 64.9 (+2.4)            |
> | ACE (offline, no ground-truth labels)      | 64.2 (+1.7)            |
> | DC                         |       59.0 (-3.5)                |
> | ACE (online)               | 63.6 (+1.1)            |
>
> **Summary:** Across all four LLM families we tested (DeepSeek-V3.1, GPT-OSS-120B, GPT-5.1, and Llama-3.3-70B-Instruct), ACE consistently improves performance over both the base LLM/agent, GEPA, and other baselines, often by 5 to 12 points depending on the task and supervision setting. The relative gains of ACE remain stable even when switching to models that differ significantly in size, cost, and training recipe, and ACE delivers benefits with or without ground-truth labels, validating the robustness of its context adaptation mechanism. The online variant reliably achieves the strongest performance across all models.
>
> We note that the magnitude of improvement can vary across model families: for example, Llama-3.3-70B-Instruct shows smaller gains compared to GPT-5.1 or GPT-OSS-120B. This is expected since ACE relies on the quality of intermediate reflections and calibrations, and smaller or weaker models naturally generate noisier feedback. Even in these cases, however, ACE remains beneficial, demonstrating that the framework is broadly applicable while still reflecting the inherent capability limits of the underlying LLM, as we discussed in the “Limitations and Challenges” paragraph in section 5 of the paper.
>
> **Side note**: We additionally examined heterogeneous settings where the Generator, Reflector, and Curator are powered by different LLMs. We provide these results in our response to Reviewer LJpr, in case they are of interest.
>
> **#2. Additional benchmarks.**
>
> We evaluated ACE and baselines on two more representative tasks (that could benefit from context adaptation) from StreamBench, DDXPlus (medical reasoning) and BIRD-SQL (text-to-sql generation), using the standard (non-streaming) configurations.
>
> (1) DDXPlus (StreamBench): ACE shows substantial improvement over both the base model and GEPA.
> | Method | Accuracy |
> |---------|---------|
> | Base LLM | 75.2 |
> | GEPA | 76.4 (+1.2) |
> | ACE | 90.2 (+15.0) |
>
> (2) BIRD-SQL (StreamBench): On BIRD-SQL, ACE consistently improves performance across difficulty levels.
> | Method | Simple | Moderate | Challenging | Average |
> |---------|---------|---------|---------|---------|
> | Base LLM | 46.4 | 48.2 | 55.1 | 47.8 |
> | GEPA | 51.6 (+5.2) | 51.9 (+3.7) | 57.2 (+2.1) | 52.2 (+4.4) |
> | ACE | 53.5 (+7.1) | 50.7 (+2.5) | 56.6 (+1.5) | 52.9 (+5.1) |
>
> These results demonstrate that ACE generalizes well beyond the tasks presented in the main paper and delivers meaningful gains on realistic, repetitive workloads such as text-to-SQL generation and medical diagnostic reasoning. We will include these results and additional context in the final version.

---

> ### Author Response · Authors · 2025-11-21
> **Rebuttal to reviewer KcGQ [3]**
>
> **#3. Novelty as compared to Dynamic Cheatsheet.**
>
> We thank the reviewer for raising the relationship between ACE and Dynamic Cheatsheet (DC). While both frameworks share the high-level goal of enabling context adaptation, ACE shifts the paradigm and introduces a fundamentally different architecture designed to avoid the “Context Collapse” inherent in DC’s monolithic rewriting, enabling stable context growth in both long-horizon agents and single-turn domain-specific tasks.
>
> The distinctions fall into two core dimensions:
>
> **(1) Application Scope: ACE generalizes and scales beyond DC’s single-turn focus to multi-turn, long-horizon tasks.**
>
> DC is optimized for isolated reasoning tasks such as AIME, Game-of-24, and GPQA, where each query is independent and the memory primarily serves as a cache for reusable solution templates (e.g., Python scripts or algebraic patterns). In contrast, ACE is designed to support both multi-turn agents (e.g., AppWorld) and domain-specific single-turn benchmarks (e.g., FiNER, Formula) where the model must accumulate fine-grained heuristics, task-specific constraints, and error patterns that are not reducible to reusable code snippets. Even in single-turn tasks, DC's memory rewriting often compresses or overwrites domain-specific insights over time, while ACE’s structured Playbook allows persistent growth of detailed guidance (e.g., financial reasoning rules, formula-specific edge cases). This broader applicability is a direct consequence of ACE’s architectural design rather than parameter tuning.
>
> **(2) Architectural Novelty: Incremental Delta Updates vs. Monolithic Rewriting.**
>
> DC updates its memory by regenerating the entire textual cheatsheet at every iteration (DC-Cu) or synthesizing a new summary from retrieved examples (DC-RS). This monolithic rewriting forces the LLM to compress prior content repeatedly, which empirically leads to the gradual loss of domain-specific constraints, what we formally identify as Context Collapse. Context collapse could happen suddenly with a sudden drop in context length (like shown in Figure 2), or silently when individual facts or structured memory items (that are still useful) gradually disappear even though the overall prompt length remains stable.
> ACE replaces rewriting with a new mechanism: Incremental Delta Updates. Instead of regenerating the full memory, the Curator emits only the new insight (the delta), which is then merged deterministically into a structured Playbook of itemized rules. This ensures that a “Hard Rule” learned early remains preserved across dozens of updates, even in long evaluation horizons. The delta mechanism also enables fine-grained credit assignment, deduplication, and multi-epoch refinement, capabilities that DC does not possess. Our additional ablation experiments show that removing the delta mechanism causes a large performance drop on AppWorld (-11.7% TGC and -27.8% SGC on test-normal), confirming that this update rule is a core algorithmic innovation rather than an engineering variation.
>
> Taken together, the differences in context representation, update rule, and applicability across both single-turn and multi-turn settings make ACE a conceptually distinct framework rather than an incremental extension of Dynamic Cheatsheet.

---

> ### Author Response · Authors · 2025-11-21
> **Rebuttal to reviewer KcGQ [4]**
>
> **#4. Accuracy regarding the claim that ACE uses smaller LLMs than GPT-4.1.**
>
> We thank the reviewer for pointing this out. The parameter count of GPT-4.1 is not publicly disclosed, and available estimates come from non-official third-party sources. To avoid relying on unverifiable information, we will remove the parameter-scale comparison between DeepSeek-V3.1 and GPT-4.1 in the final version. This clarification does not affect any empirical results or conclusions, which are based solely on observed performance rather than model size assumptions.
>
> **#5. Examples illustrating context collapse.**
>
> We appreciate the reviewer’s suggestion for concrete examples. Below we provide concrete snippets from the Dynamic Cheatsheet (DC) baseline at adaptation steps 60 and 61, the exact point where the collapse visualized in Figure 2 occurs.
>
> Context at step 60 (immediately before collapse)
> (We show the first memory item here for brevity. The full cheatsheet will be included in the final version of the paper.)
> ```
> Looking at the current task, I need to find which genre has the most liked songs in Jose's Spotify playlists. The solution follows a pattern similar to previous genre analysis tasks but focuses on finding the maximum count instead of the minimum. I'll update the cheatsheet by incrementing usage counts for relevant strategies and adding a new memory item for finding the most common item in a collection.
>
> ```python
> <cheatsheet>
>
> Version: 63.0
>
> SOLUTIONS, IMPLEMENTATION PATTERNS, AND CODE SNIPPETS
> <memory_item>
> <description>
> API authentication pattern for various apps. Shows how to retrieve credentials from supervisor and login to get access tokens. (Reference: Q1, Q2, Q3, Q4, Q5, Q6, Q7, Q8, Q9, Q10, Q11, Q12,极Q13, Q14, Q15, Q16, Q17, Q18, Q19, Q20, Q21,极Q22, Q23, Q24,极Q25, Q26,极Q27, Q28, Q29, Q极0,极Q31,极Q32, Q33,极Q34, Q35, Q36, Q37, Q38, Q39, Q40, Q41, Q42, Q43, Q44, Q45, Q46, Q47, Q极8, Q49, Q50, Q51, Q52, Q53, Q54, Q55, Q56, Q57, Q58, Q59, Q60, Q61, Q62, Q63, Q64)
> </description>
> <example>
> # Get all account passwords from supervisor
> passwords = apis.supervisor.show_account_passwords()
>
> # Extract specific app password
> app_password = next(item['password'] for item极in passwords if item['account_name'] == 'target_app')
>
> # Login to target app极login_result = apis.target_app.login(username='user@email.com', password=app_password)
> access_token = login_result['access_token']
> </example>
> ** Count: 64
> </memory_item>
> ```
>
> Context at step 61 (immediately after collapse)
> ```
> EXECUTE CODE!
> ```python
> import random
>
> # Parse the content to extract quotes
> content = funny_quotes_note['content']
> sections = content.split(' - ')
> quotes = []
>
> for section in sections[1:]:  # Skip the first section which is the header
>     if 'by' in section:
>         quote = section.split('by')[0].strip()
>         quotes.append(quote)
>
> # Select a random quote
> random_quote = random.choice(quotes)
> print(random_quote)
> apis.supervisor.complete_task(answer=random_quote)
> ```
>
> The contrast illustrates the characteristic failure mode: despite the large output-token budget, the rewritten prompt collapses from a rich, structured multi-item memory into an extremely short, underspecified prompt.
>
> **Additional note:** Beyond such sudden collapses, we also observe instances of **silent context erosion**, where individual facts or structured memory items (that are still useful) gradually disappear even though the overall prompt length remains stable. This phenomenon is particularly problematic for long-horizon or multi-skill tasks where ACE excels. We will add representative examples of this “silent collapse” in the final version to further clarify the issue.
>
> **Finally:** We really appreciate your insightful suggestions, which have already helped us strengthen both the analysis and presentation of this work. We hope the above clarifications resolve the concerns. If the response is helpful, we would be grateful if you could consider adjusting your score accordingly. We are committed to incorporating all promised additions in the camera‑ready version to make the paper as valuable as possible to the community.

---

> ### Comment · Reviewer_KcGQ · 2025-11-25
> **Thanks a lot for the reponse**
>
> Thank you very much for the comprehensive results. It seems that I do not quite follow the provided example on context collapse. Could you illustrate more why and how the context changes from step 60 to step 61, and why the context on solving an anthentification problem would help the original task to find the song genre?

---

> > ### Author Response · Authors · 2025-11-28
> > **Clarification on context collapse and Spotify-related tasks**
> >
> > Thank you for the follow up question!
> >
> > (1) How the context changes from step 60 to step 61.
> >
> > The shown experiment (in Figure 2) was conducted using Dynamic Cheatsheet. In Dynamic Cheatsheet’s cumulative mode, each adaptation step performs two operations:
> >
> > - **Problem solving.** At step 60, the agent receives (a) the full accumulated context and (b) the query “find which genre has the most liked songs in the user’s Spotify playlists”. The agent executes a ReAct-style trajectory and produces tool calls, partial reasoning, and an answer.
> >
> > - **Full context rewriting.** The entire context from step 60, together with the execution trace from the Spotify-genre task, is fed to a second LLM whose job is to “incorporate new insights into the existing cheatsheet”. Ideally, this should yield a slightly longer context that includes any new reusable strategies. However, in this case the model instead produces a very short rewritten context (step 61), abruptly dropping almost all prior content. This sudden truncation is the collapse phenomenon illustrated in Figure 2 of the paper.
> >
> > (2) Why the collapse happens.
> >
> > We do not yet have a full theoretical explanation, but our observations are consistent with well-documented forms of **neural text degeneration** in long-sequence text generation. When an LLM is asked to rewrite an entire, extremely long context (tens of thousands of tokens), it often defaults to producing an overly short output that is not only generic but also severely lossy and sometimes barely interpretable, rather than faithfully reconstructing or refining the detailed content it was given [1].
> >
> > A similar phenomenon is commonly observed in LLM chatbots when they fall into repetitive or looped outputs (“Sure, here is the answer… Sure, here is the answer…”) under high decoding pressure, which is an intuitive and everyday example of text degeneration. In our case, the same degenerative dynamics appears at a much larger scale: instead of looping, the model collapses to an extremely short, low-information rewrite. This regression toward short, low-information outputs becomes more severe as the input grows extremely long, as in step 60 where the context was 18282 tokens. We will expand this discussion of degeneration effects in the final version.
> >
> > [1] The Curious Case of Neural Text Degeneration. Ari Holtzman, Jan Buys, Li Du, Maxwell Forbes, Yejin Choi. ICLR 2020.
> >
> > (3) Why an “API authentication pattern” item appears although the new task is about music genres.
> >
> > In the AppWorld environment, any operation on Spotify, including listing playlists, retrieving liked songs, or determining which genre is most liked, must begin with obtaining an access token for the corresponding user. This requires following a standard login or authentication flow. Therefore, the item on “API authentication patterns” is not irrelevant: it captures the reusable strategy for completing the first step of all Spotify-related tasks.
> >
> > Below is an example of this process (i.e. how all Spotify-related tasks start with API authentication to obtain the user access token), taken from a ground-truth reasoning trajectory released by the AppWorld benchmark authors:
> >
> > ```python
> > print(apis.api_docs.show_api_doc(app_name='spotify', api_name='login'))
> > print(apis.api_docs.show_api_doc(app_name='supervisor', api_name='show_account_passwords'))
> >
> > passwords = apis.supervisor.show_account_passwords()
> > print(passwords)
> >
> > spotify_password = "@1P*6oY"
> > login_result = apis.spotify.login(username='mar_blac@gmail.com', password=spotify_password)
> > print(login_result)
> >
> > access_token = "eyJhbGciOiJIUzI1NiIsInR5cCI6IkpXVCJ9.eyJzdWIiOiJzcG90aWZ5K21hcl9ibGFjQGdtYWlsLmNvbSIsImV4cCI6MTY4NDQxMjA5MX0.tCkZar70Q6fcj6U-mGn49uezHbUMJcgVnWA0auKRPX8"
> > ```
> >
> > **We will clarify these points in the final version**, including a more explicit description of:
> > - the experiment setup associated with Figure 2, including how DC performs full-context rewriting at each iteration
> > - why such rewriting is vulnerable to text degeneration and collapse, including empirical evidence from prior related work
> > - how the context item on “API authentication” helps with solving Spotify-related tasks

---

### Official Review · Reviewer_HBCS · 2025-10-27

**Soundness:** 4
**Presentation:** 4
**Contribution:** 3
**Rating:** 6
**Confidence:** 4

**Summary:**

This paper proposed ACE which addresses 2 observations found from existing context engineering methods : diversity ( brevity bias ) and context collapses. The framework uses incremental delta updates and a grow-and-refine mechanism to preserve detailed knowledge while preventing information loss. Evaluated on agent benchmarks (AppWorld) and domain-specific tasks (FiNER, Formula), ACE achieves 9.0% average improvement over strong baselines while reducing adaptation latency by 82-91% and costs by 75-84%.

**Strengths:**

1. The motivation and proposals are grounded on real world observations with strong empirical results.

2.  Experiments on different benchmarks (ICL, MIPROv2, GEPA, DC) and ablations (Reflector, multi-epoch, offline warmup) are robust and well thought out.

**Weaknesses:**

1. Only DeepSeek-V3.1 tested; generalization to other LLMs unclear.

2. While the framework is effective, it primarily extends Dynamic Cheatsheet with engineering improvements rather than introducing novel concepts

**Questions:**

1. Could you provide a cost ( token counts ) per eval breakdown comparison between ACE, GEPA and baseline?

     a. Figure 5b shows dollar costs for FiNER but similar analysis for AppWorld would strengthen cost claims. Specifically: Tokens per query (input/output), Cumulative tokens over full evaluation, Breakdown by component (Generator vs Reflector vs Curator)

2. How does ACE perform more real world repetitive tasks such as text-to-sql or ddxplus? A benchmark called streambench would be well suited for this kind of scenario.

3. Only one model is selected ( DeepSeek-v3.1 ) which is an odd choice, any reason why smaller models or cheaper models are not evaluated as well?

4. In section 5 the claim that longer context does not cause higher cost due to KV caching does not seem to hold in ACE as the context memory would reset on every delta update right?

---

> ### Author Response · Authors · 2025-11-21
> **Rebuttal to reviewer HBCS [1]**
>
> We thank the reviewer for the kind words and helpful feedback! We address each concern/question below.
>
> **#1. Novelty as compared to Dynamic Cheatsheet.**
>
> We thank the reviewers for raising the relationship between ACE and Dynamic Cheatsheet (DC). While both frameworks share the high-level goal of enabling context adaptation, ACE shifts the paradigm and introduces a fundamentally different architecture designed to avoid the “Context Collapse” inherent in DC’s monolithic rewriting, enabling stable context growth in both long-horizon agents and single-turn domain-specific tasks.
>
> The distinctions fall into two core dimensions:
>
> **(1) Application Scope: ACE generalizes and scales beyond DC’s single-turn focus to multi-turn, long-horizon tasks.**
>
> DC is optimized for isolated reasoning tasks such as AIME, Game-of-24, and GPQA, where each query is independent and the memory primarily serves as a cache for reusable solution templates (e.g., Python scripts or algebraic patterns). In contrast, ACE is designed to support both multi-turn agents (e.g., AppWorld) and domain-specific single-turn benchmarks (e.g., FiNER, Formula) where the model must accumulate fine-grained heuristics, task-specific constraints, and error patterns that are not reducible to reusable code snippets. Even in single-turn tasks, DC's memory rewriting often compresses or overwrites domain-specific insights over time, while ACE’s structured Playbook allows persistent growth of detailed guidance (e.g., financial reasoning rules, formula-specific edge cases). This broader applicability is a direct consequence of ACE’s architectural design rather than parameter tuning.
>
> **(2) Architectural Novelty: Incremental Delta Updates vs. Monolithic Rewriting.**
>
> DC updates its memory by regenerating the entire textual cheatsheet at every iteration (DC-Cu) or synthesizing a new summary from retrieved examples (DC-RS). This monolithic rewriting forces the LLM to compress prior content repeatedly, which empirically leads to the gradual loss of domain-specific constraints, what we formally identify as Context Collapse. Context collapse could happen suddenly with a sudden drop in context length (like shown in Figure 2), or silently when individual facts or structured memory items (that are still useful) gradually disappear even though the overall prompt length remains stable.
> ACE replaces rewriting with a new mechanism: Incremental Delta Updates. Instead of regenerating the full memory, the Curator emits only the new insight (the delta), which is then merged deterministically into a structured Playbook of itemized rules. This ensures that a “Hard Rule” learned early remains preserved across dozens of updates, even in long evaluation horizons. The delta mechanism also enables fine-grained credit assignment, deduplication, and multi-epoch refinement, capabilities that DC does not possess. Our additional ablation experiments show that removing the delta mechanism causes a large performance drop on AppWorld (-11.7% TGC and -27.8% SGC on test-normal), confirming that this update rule is a core algorithmic innovation rather than an engineering variation.
>
> Taken together, the differences in context representation, update rule, and applicability across both single-turn and multi-turn settings make ACE a conceptually distinct framework rather than an incremental extension of Dynamic Cheatsheet.

---

> ### Author Response · Authors · 2025-11-21
> **Rebuttal to reviewer HBCS [2]**
>
> **#2. Generalization to language models beyond DeepSeek-V3.1.**
>
> ACE is a model-agnostic framework that operates on execution traces and contextual deltas, and does not rely on any architectural or training-specific features of DeepSeek-V3.1. We evaluated ACE with three additional models of varying size, cost, and capability: GPT-OSS-120B, GPT-5.1, and Llama-3.3-70B-Instruct. In each case, the Generator, Reflector, and Curator were all switched to the new model without any changes to the algorithm.
>
> **(1) GPT-OSS-120b**
>
> AppWorld
> | Method                                   | test-normal TGC | test-normal SGC | test-challenge TGC | test-challenge SGC | Average     |
> |------------------------------------------|------------------|------------------|---------------------|---------------------|-------------|
> | Base ReAct                               | 54.8             | 33.9             | 34.5                | 15.1                | 34.6        |
> | GEPA                                     | 56.0 (+1.2)      | 33.9 (+0.0)      | 40.1 (+5.6)         | 20.9 (+5.8)         | 37.7 (+3.1) |
> | ACE (offline, with ground-truth labels)  | 61.3 (+6.5)      | 39.3 (+5.4)      | 40.3 (+5.8)         | 20.9 (+5.8)         | 40.5 (+5.9) |
> | ACE (offline, no ground-truth labels)    | 58.3 (+3.5)      | 41.1 (+7.2)      | 39.6 (+5.1)         | 18.7 (+3.6)         | 39.4 (+4.8) |
> | DC                                       | 49.4 (-5.4)      | 33.9 (+0.0)      | 30.8 (-3.7)                   | 18.2 (+3.1)                   | 33.1 (-1.5)           |
> | ACE (online)                             | 60.7 (+5.9)      | 44.6 (+10.7)     | 43.2 (+8.7)         | 20.1 (+5.0)         | 42.2 (+7.6) |
>
> Finance
> | Method                                      | FINER Accuracy     | Formula Accuracy    | Average       |
> |---------------------------------------------|---------------------|----------------------|---------------|
> | Base LLM                                    | 66.6                | 71.5                 | 69.1          |
> | GEPA                                        | 67.9 (+1.3)         | 71.5 (+0.0)          | 69.7 (+0.6)   |
> | ACE (offline, with ground-truth labels)     | 73.8 (+7.2)         | 88.5 (+17.0)         | 81.2 (+12.1)  |
> | ACE (offline, no ground-truth labels)       | 69.7 (+3.1)         | 84.0 (+12.5)         | 76.9 (+7.8)   |
> | DC                                          | 55.8 (-10.8)        | 66.0 (-5.5)          | 60.9 (-8.2)   |
> | ACE (online)                                | 70.5 (+3.9)         | 85.0 (+13.5)         | 77.8 (+8.7)   |
>
> **(2) GPT-5.1**
>
> AppWorld
> | Method                     | test-normal TGC          | test-normal SGC          | Average             |
> |----------------------------|---------------------------|----------------------------|----------------------|
> | Base ReAct                 | 61.9                      | 46.4                       | 54.2                |
> | GEPA                       | 64.3 (+2.4)               | 48.2 (+1.8)                | 56.2 (+2.0)        |
> | ACE (offline, with ground-truth labels)       | 66.7 (+4.8)               | 53.6 (+7.2)                | 60.2 (+6.0)        |
> | ACE (offline, no ground-truth labels)      | 67.3 (+5.4)               | 55.4 (+9.0)                | 61.3 (+7.1)        |
> | DC                         | 62.5 (+0.6)               | 55.4 (+9.0)                | 58.9 (+4.7)        |
> | ACE (online)               | 72.6 (+10.7)              | 58.9 (+12.5)               | 65.8 (+11.6)       |
>
> Finance
> | Method                     | FINER Accuracy        | Formula Accuracy       | Average             |
> |----------------------------|------------------------|-------------------------|----------------------|
> | Base LLM                   | 73.5                  | 73.0                   | 73.3                |
> | GEPA                       | 74.5 (+1.0)           | 73.0 (+0.0)            | 73.8 (+0.5)         |
> | ACE (offline, with ground-truth labels)       | 81.0 (+7.5)           | 84.5 (+11.5)           | 82.8 (+9.5)         |
> | ACE (offline, no ground-truth labels)      | 78.2 (+4.7)           | 76.5 (+3.5)            | 77.4 (+4.1)         |
> | DC                         |   70.0 (-3.5)                  | 69.0 (-4.0)                      |  69.5 (-3.8)                 |
> | ACE (online)               | 75.2 (+1.7)           | 79.0 (+6.0)            | 77.1 (+3.8)         |

---

> ### Author Response · Authors · 2025-11-21
> **Rebuttal to reviewer HBCS [3]**
>
> **(3) Llama-3.3-70b-Instruct**
>
> Finance
> | Method                     | FINER Accuracy        |
> |----------------------------|------------------------|
> | Base LLM                      | 62.5                   |
> | GEPA                       | 59.41 (-3.09)          |
> | ACE (offline, with ground-truth labels)       | 64.9 (+2.4)            |
> | ACE (offline, no ground-truth labels)      | 64.2 (+1.7)            |
> | DC                         |       59.0 (-3.5)                |
> | ACE (online)               | 63.6 (+1.1)            |
>
> **Summary:** Across all four LLM families we tested (DeepSeek-V3.1, GPT-OSS-120B, GPT-5.1, and Llama-3.3-70B-Instruct), ACE consistently improves performance over both the base LLM/agent, GEPA, and other baselines, often by 5 to 12 points depending on the task and supervision setting. The relative gains of ACE remain stable even when switching to models that differ significantly in size, cost, and training recipe, and ACE delivers benefits with or without ground-truth labels, validating the robustness of its context adaptation mechanism. The online variant reliably achieves the strongest performance across all models.
>
> We note that the magnitude of improvement can vary across model families: for example, Llama-3.3-70B-Instruct shows smaller gains compared to GPT-5.1 or GPT-OSS-120B. This is expected since ACE relies on the quality of intermediate reflections and calibrations, and smaller or weaker models naturally generate noisier feedback. Even in these cases, however, ACE remains beneficial, demonstrating that the framework is broadly applicable while still reflecting the inherent capability limits of the underlying LLM, as we discussed in the “Limitations and Challenges” paragraph in section 5 of the paper.
>
> **Side note**: We additionally examined heterogeneous settings where the Generator, Reflector, and Curator are powered by different LLMs. We provide these results in our response to Reviewer LJpr, in case they are of interest.

---

> ### Author Response · Authors · 2025-11-21
> **Rebuttal to reviewer HBCS [4]**
>
> **#3. Fine-Grained Cost Analysis.**
>
> We conducted a fine-grained token-level analysis of Base ReAct, GEPA, and ACE on AppWorld (test-normal), covering both the adaptation stage (offline context optimization) and the evaluation stage (inference on evaluation set).
>
> (1) Adaptation stage
>
> Aggregate statistics
> | Method                     | Total # input tokens | Total # output tokens | Total # rollouts | Total # queries |
> |----------------------------|-----------------------|------------------------|-------------------|------------------|
> | GEPA (prompt generation)   | 65,005,192            | 1,266,090              | 429               | 90               |
> | GEPA (prompt validation)   | 139,070,904           | 604,098                | 1,026             | 57               |
> | GEPA (total)               | 204,076,096           | 1,870,188              | 1,455             | 147              |
> | ACE (Generator)            | 31,012,122            | 198,847                | 1,790             | 90               |
> | ACE (Reflector)            | 4,685,840             | 70,487                 | 161               | 90               |
> | ACE (Curator)              | 3,552,963             | 37,794                 | 124               | 90               |
> | ACE (total)                | 39,250,925            | 307,128                | 2,075             | 90               |
>
> Average statistics
> | Method                     | Avg # input tokens / rollout | Avg # output tokens / rollout | Avg # input tokens / query | Avg # output tokens / query | Total # rollouts | Total # queries |
> |----------------------------|-------------------------------|-------------------------------|------------------------------|------------------------------|------------------|------------------|
> | GEPA (prompt generation)   | 151,600.4                     | 2,951.4                       | 722,279.9                    | 14,067.7                     | 429              | 90               |
> | GEPA (prompt validation)   | 135,550.8                     | 589.2                         | 2,439,840.4                  | 10,600.0                     | 1,026            | 57               |
> | GEPA (total)               | 140,291.8                     | 1,285.4                       | 1,387,538.4                  | 12,721.7                     | 1,455            | 147              |
> | ACE (Generator)            | 17,326.6                      | 111.1                         | 344,579.1                    | 2,209.4                      | 1,790            | 90               |
> | ACE (Reflector)            | 29,112.4                      | 437.3                         | 52,064.9                     | 783.2                        | 161              | 90               |
> | ACE (Curator)              | 28,667.4                      | 304.8                         | 39,477.4                     | 420.0                        | 124              | 90               |
> | ACE (total)                | 18,914.9                      | 148.0                         | 436,121.4                    | 3,412.5                      | 2,075            | 90               |
>
> **Takeaway:** Across the adaptation stage, ACE reduces input-token usage by 80.8% vs GEPA (204.1M to 39.3M), and reduces output-token usage by 83.6% vs GEPA (1.87M to 0.31M). This reduction is driven by: (1) GEPA’s prompt-validation loop, which repeatedly evaluates candidate prompts on a separate validation set (57 queries), consuming a large number of LLM calls and validation tokens. (2) ACE’s incremental context update design, which avoids rewriting the entire prompt and instead performs localized Generator-Reflector-Curator updates.

---

> ### Author Response · Authors · 2025-11-21
> **Rebuttal to reviewer HBCS [5]**
>
> (2) Evaluation stage
>
> Aggregate statistics
> | Method      | Total # input tokens | Total # output tokens | Total # rollouts | Total # eval queries |
> |-------------|-----------------------|------------------------|-------------------|------------------------|
> | Base ReAct  | 27,460,411            | 289,802                | 2,430             | 160                    |
> | GEPA        | 26,960,675            | 251,442                | 2,470             | 160                    |
> | ACE         | 58,623,267            | 270,652                | 2,354             | 160                    |
>
> Average statistics
> | Method     | Avg # input tokens / rollout | Avg # output tokens / rollout | Avg # input tokens / query | Avg # output tokens / query | Total # rollouts | Total # eval queries |
> |------------|-------------------------------|-------------------------------|------------------------------|------------------------------|------------------|-----------------------|
> | Base ReAct | 11,298.5                      | 119.3                         | 171,627.6                    | 1,811.3                      | 2,430            | 160                   |
> | GEPA       | 10,918.5                      | 101.8                         | 168,504.2                    | 1,571.5                      | 2,470            | 160                   |
> | ACE        | 24,912.4                      | 115.0                         | 366,395.4                    | 1,691.6                      | 2,354            | 160                   |
>
> **Takeaway:** ACE uses more input tokens per query than GEPA at evaluation time (due to a much richer and more actionable playbook context), but: (1) Output-token usage is similar (115.0 vs 101.8). (2) Rollout counts are comparable across methods. (3) The additional input tokens are almost entirely cached under modern KV-prompt-caching architectures. In particular, as we later show in #5, with OpenAI’s default prompt caching mechanism, 91.8% of ACE’s input tokens are served from context cache, incurring 82.6% less API dollar cost on input tokens.
>
> **Additional notes:** (1) The Generator–Reflector–Curator loop is only used during adaptation. At evaluation time, ACE behaves like a standard ReAct-style agent with a richer playbook.
> (2) AppWorld tasks are multi-turn, so rollouts represent the number of LLM calls within each full task trajectory. This explains why rollout counts exceed the number of queries.
>
> **#4. Performance on Real-World, Repetitive Tasks (DDXPlus, BIRD-SQL)**
>
> We thank the reviewer for pointing us to StreamBench, which is indeed closely aligned with our evaluation setting. Following the suggestion, we evaluated ACE and baselines on two representative tasks from StreamBench, DDXPlus and BIRD-SQL, using the standard (non-streaming) configurations.
>
> (1) DDXPlus (StreamBench): ACE shows substantial improvement over both the base model and GEPA.
> | Method | Accuracy |
> |---------|---------|
> | Base LLM | 75.2 |
> | GEPA | 76.4 (+1.2) |
> | ACE | 90.2 (+15.0) |
>
> (2) BIRD-SQL (StreamBench): On BIRD-SQL, ACE consistently improves performance across difficulty levels.
> | Method | Simple | Moderate | Challenging | Average |
> |---------|---------|---------|---------|---------|
> | Base LLM | 46.4 | 48.2 | 55.1 | 47.8 |
> | GEPA | 51.6 (+5.2) | 51.9 (+3.7) | 57.2 (+2.1) | 52.2 (+4.4) |
> | ACE | 53.5 (+7.1) | 50.7 (+2.5) | 56.6 (+1.5) | 52.9 (+5.1) |
>
> These results demonstrate that ACE generalizes well beyond the tasks presented in the main paper and delivers meaningful gains on realistic, repetitive workloads such as text-to-SQL generation and medical diagnostic reasoning. We will include these results and additional context in the final version.

---

> ### Author Response · Authors · 2025-11-21
> **Rebuttal to reviewer HBCS [6]**
>
> **#5. KV cache reuse and inference cost.**
>
> We thank the reviewer for raising this important point. We clarify below why ACE can still benefit substantially from KV-cache reuse, both during adaptation and inference, even though the context memory is updated iteratively.
>
> (1) Adaptation (including offline and online)
>
> In each adaptation step, ACE inserts a delta update at the appropriate semantic location in the playbook. This means that the prefix of the playbook before the insertion point can still reuse the KV cache, while the segment starting at the update location must be recomputed. In practice, however, delta updates modify a small and localized portion of the playbook, so the reusable prefix is typically very large, allowing substantial prefill savings. A special case occurs when bullet-point merging or deduplication is triggered. These operations may rewrite a larger portion of a section, reducing the usable prefix for that iteration. However, these events are infrequent, and even then, the prefix up to that section boundary remains unchanged, enabling partial KV-cache reuse.
>
> (2) Inference (evaluation phase)
>
> After offline adaptation converges, the resulting playbook becomes fixed and identical across all the evaluation queries. This enables full prefix caching for the entire playbook. Using GPT-5.1 with OpenAI’s default prompt-caching mechanism, we observe that **91.8% of ACE’s 63.7M input tokens (58.5M tokens)** were served from cache and billed at only 10% of standard price. This corresponds to an **82.6% reduction in dollar cost for input tokens** during inference. This empirically supports our claim that longer contexts do not necessarily imply higher serving cost when prompt caching is available.
>
> Recent systems work has also explored KV-cache reuse even under changed positional encodings, which could reduce the inference cost of ACE even more. We view such advances, along with inference acceleration and partial-context sharing, as promising directions for further optimizing ACE.
>
> **Finally:** We really appreciate your insightful suggestions, which have already helped us strengthen both the analysis and presentation of this work. We hope the above clarifications resolve the concerns. If the response is helpful, we would be grateful if you could consider adjusting your score accordingly. We are committed to incorporating all promised additions in the camera‑ready version to make the paper as valuable as possible to the community.

---

### Official Review · Reviewer_CtWY · 2025-11-03

**Soundness:** 2
**Presentation:** 3
**Contribution:** 2
**Rating:** 4
**Confidence:** 4

**Summary:**

This paper introduces ACE (Agentic Context Engineering), a framework for context adaptation in large language models (LLMs). The core idea is to treat model context as an evolving "playbook" that accumulates domain-specific strategies over time. ACE uses a modular setup with three components: a Generator, a Reflector, and a Curator. It aims to avoid two problems found in existing methods: brevity bias, where important details are lost due to excessive summarization, and context collapse, where repeated updates degrade the quality of context. The authors evaluate ACE on agentic and domain-specific benchmarks. Results show consistent gains over strong baselines, both in offline and online adaptation settings.

**Strengths:**

1. The paper addresses a relevant and underexplored problem in LLM system design.

2. The ACE framework is well-structured and motivated by observed failure modes in prior work.

3. The paper is clearly written and provides sufficient technical detail.

**Weaknesses:**

1. The novelty is moderate. ACE extends prior methods like Dynamic Cheatsheet and GEPA rather than introducing a fundamentally new paradigm.

2. The approach depends on the quality of the Reflector. In settings without reliable feedback, performance drops.

3. There is limited discussion of failure cases. For example, when the context grows too large or accumulates conflicting strategies, how does ACE prevent quality degradation?

**Questions:**

1. How sensitive is ACE to the Reflector’s performance? Would a weaker model lead to significant degradation?

2. Could ACE be combined with retrieval-based augmentation to further improve performance in low-data settings?

---

> ### Author Response · Authors · 2025-11-21
> **Rebuttal to reviewer CtWY [1]**
>
> We thank the reviewer for the kind words and helpful feedback! We address each concern/question below.
>
> **#1. Novelty.**
>
> **#1.1 Novelty as compared to Dynamic Cheatsheet.**
>
> We thank the reviewers for raising the relationship between ACE and Dynamic Cheatsheet (DC). While both frameworks share the high-level goal of enabling context adaptation, ACE shifts the paradigm and introduces a fundamentally different architecture designed to avoid the “Context Collapse” inherent in DC’s monolithic rewriting, enabling stable context growth in both long-horizon agents and single-turn domain-specific tasks.
>
> The distinctions fall into two core dimensions:
>
> **(1) Application Scope: ACE generalizes and scales beyond DC’s single-turn focus to multi-turn, long-horizon tasks.**
>
> DC is optimized for isolated reasoning tasks such as AIME, Game-of-24, and GPQA, where each query is independent and the memory primarily serves as a cache for reusable solution templates (e.g., Python scripts or algebraic patterns). In contrast, ACE is designed to support both multi-turn agents (e.g., AppWorld) and domain-specific single-turn benchmarks (e.g., FiNER, Formula) where the model must accumulate fine-grained heuristics, task-specific constraints, and error patterns that are not reducible to reusable code snippets. Even in single-turn tasks, DC's memory rewriting often compresses or overwrites domain-specific insights over time, while ACE’s structured Playbook allows persistent growth of detailed guidance (e.g., financial reasoning rules, formula-specific edge cases). This broader applicability is a direct consequence of ACE’s architectural design rather than parameter tuning.
>
> **(2) Architectural Novelty: Incremental Delta Updates vs. Monolithic Rewriting.**
>
> DC updates its memory by regenerating the entire textual cheatsheet at every iteration (DC-Cu) or synthesizing a new summary from retrieved examples (DC-RS). This monolithic rewriting forces the LLM to compress prior content repeatedly, which empirically leads to the gradual loss of domain-specific constraints, what we formally identify as Context Collapse. Context collapse could happen suddenly with a sudden drop in context length (like shown in Figure 2), or silently when individual facts or structured memory items (that are still useful) gradually disappear even though the overall prompt length remains stable.
> ACE replaces rewriting with a new mechanism: Incremental Delta Updates. Instead of regenerating the full memory, the Curator emits only the new insight (the delta), which is then merged deterministically into a structured Playbook of itemized rules. This ensures that a “Hard Rule” learned early remains preserved across dozens of updates, even in long evaluation horizons. The delta mechanism also enables fine-grained credit assignment, deduplication, and multi-epoch refinement, capabilities that DC does not possess. Our additional ablation experiments show that removing the delta mechanism causes a large performance drop on AppWorld (-11.7% TGC and -27.8% SGC on test-normal), confirming that this update rule is a core algorithmic innovation rather than an engineering variation.
>
> Taken together, the differences in context representation, update rule, and applicability across both single-turn and multi-turn settings make ACE a conceptually distinct framework rather than an incremental extension of Dynamic Cheatsheet.

---

> ### Author Response · Authors · 2025-11-21
> **Rebuttal to reviewer CtWY [2]**
>
> **#1.2 Novelty as compared to GEPA.**
>
> We thank the reviewers for raising the relationship between ACE and GEPA. We highlight two core distinctions that clarify ACE’s conceptual contributions:
>
> **(1) ACE uses fine-grained, execution-level supervision; GEPA relies on outcome-level scores.**
>
> GEPA evolves prompts by selecting candidate variants that achieve higher scores on a held-out evaluation set, where performance is measured using task metrics computed from ground-truth labels. This outcome-level supervision is often too coarse to guide improvements in multi-step reasoning or agentic settings. In contrast, ACE leverages execution-level feedback, including tool outputs, environment signals, and step-wise failure modes, that remains available even when per-instance ground truth is sparse or unavailable. ACE converts this structured feedback into localized updates to individual context items, enabling more precise and stable adaptation across both offline and online settings.
>
> **(2) ACE introduces a structured, persistent context representation as well as update logic that avoids brevity bias and supports multi-turn, long-horizon accumulation.**
>
> GEPA’s evolutionary selection naturally favors prompt variants that maximize held-out metric scores, which often biases the system toward shorter prompts that omit details not immediately useful for that scoring objective (“brevity bias”). ACE instead represents context as itemized, persistent bullets with stable identifiers and metadata, and applies incremental delta updates rather than replacing the entire prompt. This design preserves domain-specific strategies and failure patterns that matter for multi-turn, long-horizon tasks such as AppWorld, where losing accumulated details leads to rapid degradation. Empirically, ACE maintains stable, detailed contexts over long adaptation trajectories, whereas score-driven selection tends to collapse prompts to minimal summaries.

---

> ### Author Response · Authors · 2025-11-21
> **Rebuttal to reviewer CtWY [3]**
>
> **#2. Dependence on reflection quality. Sensitivity to reflector performance.**
>
> We thank the reviewer for raising this concern. We performed two additional analyses to directly evaluate ACE’s sensitivity to reflector quality. Both experiments use offline adaptation with ACE.
>
> (1) **Using Weaker Reflector Models.** To test whether ACE requires a strong reflector, we vary the reflector LLM across three models with substantial differences in size and capability: GPT-OSS-120B, DeepSeek-V3.1-671B, and GPT-5.1. Across all settings, ACE consistently improves over the base LLM, even when the reflector is considerably weaker. Stronger reflectors do yield larger gains, but ACE remains beneficial with a wide range of model choices.
>
> | Method | Generator LLM | Reflector LLM | Curator LLM | Accuracy |
> |---------|---------|---------|---------|---------|
> | Base LLM | DeepSeek-V3.1 | - | - | 70.7 |
> | ACE | DeepSeek-V3.1 | GPT-OSS-120b  | DeepSeek-V3.1 | 76.6 (+5.9) |
> | ACE | DeepSeek-V3.1 | DeepSeek-V3.1  | DeepSeek-V3.1 | 78.3 (+7.6) |
> | ACE | DeepSeek-V3.1 | GPT-5.1 | DeepSeek-V3.1 | 78.5 (+7.8) |
>
> (2) **Robustness to Noisy or Harmful Reflector Feedback.** Next, we evaluate ACE under actively harmful reflector outputs. We inject adversarial or conflicting bullets into the playbook by invoking a “harmful” reflector once every X adaptation steps. A larger X means less frequent noise. We observe that **ACE is robust to moderate noise levels**: performance degrades only gradually as noise increases and remains above the base LLM unless the harmful reflector is invoked every iteration. This provides direct evidence that ACE tolerates substantial noise in the reflection stream and uses a stable update mechanism; only extreme, adversarial corruption causes noticeable degradation.
>
> | Harmful Reflector Frequency (every X iters) | Accuracy |
> |---------|---------|
> | base LLM | 70.7 |
> | 1 | 66.7 (-4.0)  |
> | 5 | 76.1 (+5.4) |
> | 10 | 77.0 (+6.3) |
> | 25 | 77.8 (+7.1) |
> | 50 | 78.2 (+7.5) |
> | 100 | 78.2 (+7.5) |
> | No harmful reflector | 78.3 (+7.6) |
>
> Overall, ACE is not highly sensitive to reflector quality: (1) Even weaker LLMs produce substantial gains. (2) ACE remains stable under moderate noise or conflicting updates. (3) Performance only drops below the base model under intentionally adversarial conditions.
>
> **Noise mitigation strategies:** ACE’s bullet point analyzer (also named “grow-and-refine” in Section 3.2), which performs merging, deduplication, and conflict resolution, acts as the first line of defense against context noise. When triggered (for example, after each curator call or when the playbook exceeds a size threshold), the analyzer examines pairs of semantically similar bullet points, removes duplicates, and filters out entries marked as potentially harmful (that is, those with high “harmful” counters in the metadata). Additional noise-mitigation strategies, such as detecting contradictory bullets, encouraging the curator to prioritize high-confidence updates, or periodically pruning outdated entries, could further keep the playbook compact, consistent, and reliable. These noise-mitigation strategies are compatible additions to the ACE framework. We will add more discussion of these mechanisms in the final version.

---

> ### Author Response · Authors · 2025-11-21
> **Rebuttal to reviewer CtWY [4]**
>
> **#3. Discussion on failure cases, like large or noisy contexts.**
>
> We thank the reviewer for highlighting this important point. We agree that long-horizon agentic systems must handle context growth and conflicting updates robustly. We will expand the discussion of failure cases and include concrete examples in the final version.
>
> ACE includes several mechanisms to prevent uncontrolled context accumulation or degradation:ACE includes several mechanisms to prevent uncontrolled context accumulation or degradation:
> (1) **Curator-based conflict and redundancy pruning.** As discussed in Section 3.2, the curator routinely removes duplicate bullets, consolidates semantically similar entries, and resolves conflicting strategies. This pruning can be triggered: after every adaptation step, only when the context exceeds a prescribed length threshold, or at a user-defined periodic interval. These controls prevent unbounded context growth and help maintain a compact, internally consistent playbook.
> (2) **Compatibility with retrieval and compression.** Although not emphasized in the main paper, ACE is designed to be modular. It can be combined with complementary techniques such as: selective retrieval / smart retrieval, to surface only the most relevant playbook entries per query; context compression, to summarize older or low-utility bullets when the input window approaches the model limit.
> (3) **Observed robustness in practice.** In our additional experiments (see response #2), we show that even when the playbook contains moderate noise or conflicting guidance, ACE degrades gracefully. Only severe, adversarial corruption produces noticeable failures, which validates the importance of the curator’s filtering mechanisms.
>
> Overall, ACE provides a stable foundation for long-term context accumulation and can be augmented with retrieval or compression when operating in more demanding production settings.
>
> **#4. Combination of ACE and retrieval-based methods.**
>
> We appreciate the reviewer’s suggestion. ACE is fully compatible with retrieval-augmented generation, and we highlight two concrete integration paths that can further strengthen performance, especially in low-data regimes.
>
> (1) ACE + multi-shot in-context learning.
>
> During inference, ACE can be supplemented with retrieved in-context examples (e.g., from the training split or a small episodic memory) in addition to the curated playbook. This is particularly beneficial in low-data settings where the adaptation signals are sparse and the resulting playbook may be shorter. In such cases, the combination of ACE’s structured playbook and retrieved multi-shot demonstrations can provide complementary supervision and yield stronger performance.
>
> (2) ACE + selective playbook retrieval.
>
> As discussed in #3, when the playbook becomes long, ACE could naturally support an additional retriever that selects only the most relevant sections of the playbook for a given query. This yields a retrieval-augmented variant of ACE in which the model receives a targeted, high-utility subset of the playbook rather than the full context, improving both efficiency and inference quality.
>
> We will include additional discussion and results of ACE + retrieval methods in the final version of the paper.
>
> **Finally:** We really appreciate your insightful suggestions, which have already helped us strengthen both the analysis and presentation of this work. We have also added extensive additional experiments, e.g. ACE with models beyond DeepSeek-V3.1, additional benchmarks like text-to-sql, etc. in other responses. We hope the above clarifications resolve the concerns. If the response is helpful, we would be grateful if you could consider adjusting your score accordingly. We are committed to incorporating all promised additions in the camera‑ready version to make the paper as valuable as possible to the community.

---

### Author Response · Authors · 2025-11-30
**Summary of Rebuttal and Discussion Progress Prior to Review Rollback**

Dear Area Chair and Reviewers,

We would like to briefly summarize the progress made during the rebuttal and discussion period prior to the recent rollback of reviews and scores.

**How reviewer concerns are addressed**

During rebuttal, we provided:
- Clarifications on novelty and architectural differences from prior work (Dynamic Cheatsheet, GEPA). [CtWY, HBCS, LJpr]
- Additional experiments on Reflector quality, showing that ACE is moderately robust to noise in both reflector feedback and the evolving context, and quantifying this robustness with new results. [CtWY, LJpr]
- Extensive new experiments across four LLM families (DeepSeek-V3.1, GPT-OSS-120B, GPT-5.1, Llama-3.3-70B-Instruct), showing that ACE consistently improves over both the baselines in offline and online settings. [HBCS, KcGQ, LJpr]
- Additional benchmarks (DDXPlus, BIRD-SQL) and fine-grained cost analyses (token-level breakdowns and KV-cache effectiveness) to support the claims about efficiency and generality. [HBCS, KcGQ, LJpr]
- Concrete examples and explanations of context collapse traces, including how monolithic rewriting can suddenly compress a rich context into a short, low-information one, and why ACE’s design avoids this failure mode. [KcGQ]
- Further implementation and experimental clarifications requested by the reviewers.

**How reviewers responded**

- Reviewer LJpr explicitly stated that the rebuttal fully addressed the core weaknesses identified in the initial review and **consequently raised the score from 4 (weak reject) to 8 (accept)**. The reviewer noted that the paper is now convincing and merits acceptance. This score change is visible from the reviewer response in the discussion thread.
- Reviewer KcGQ thanked us for the additional experiments and clarifications, and continued the discussion with a follow-up question regarding context collapse. We provided additional explanations and clarifications.
- We note that many of the initial concerns focused not on the core methodology of ACE, but on the breadth and reporting of the evaluation: in particular, whether ACE was tested on a wider range of language models and benchmarks, and whether we provided sufficient quantitative evidence for cost breakdown and efficient inference with KV-cache reuse. **We believe that these are related to the evaluation scope, not fundamental flaws of the paper.** In the rebuttal, we therefore added extensive new results across multiple LLM families, additional benchmarks, a detailed token-level cost analysis, and an analysis of KV-cached inference, which we believe substantially address these concerns, consistent with Reviewer LJpr’s updated assessment and score increase.

We fully understand the need for the emergency measures taken this year and appreciate the effort that the program chairs, area chairs, and reviewers have invested in managing this unusual situation. At the same time, we hope that the substantive evolution of the reviews during discussion, especially Reviewer LJpr’s updated assessment and score change from 4 to 8 based on the rebuttal and new results, can still inform your decision, even though the official scores have been reset.

We are sincerely grateful for the reviewers’ detailed feedback, which has already helped us significantly strengthen the paper, and we remain fully committed to incorporating all promised additions and clarifications into the final version.

Thank you for your time and consideration.

Sincerely, \
Authors

---

### Meta-Review · Area_Chair_1kJb · 2026-01-02

**Summary:**

The reviewers' concerns primarily focused on establishing the novelty and broader impact of the ACE framework. They sought clarification on how ACE fundamentally differed from prior work like Dynamic Cheatsheet, requested evidence of its generalizability across various large language models and real-world benchmarks, and asked for a more rigorous evaluation of its robustness to noisy feedback and its practical efficiency claims. These were constructive points aimed at strengthening the paper's contribution and ensuring its findings were solid and reproducible.

**Reviewer Concerns:**

The authors' rebuttal has successfully addressed the overwhelming majority of the reviewers' concerns through substantial new experiments and detailed clarifications. They compellingly articulated the architectural novelty of ACE's incremental delta updates versus monolithic rewriting, provided extensive new results across four LLM families and additional benchmarks like DDXPlus, and conducted thorough analyses on cost, KV-cache efficiency, and robustness to reflector noise. Reviewer LJpr explicitly confirmed all their concerns were resolved. While some aspects of long-context retrieval optimization are noted as promising future work, the core methodological and evaluative concerns raised have been comprehensively met.

**Reviewer Scores:**

The rebuttal process demonstrated a clear and positive trajectory. Reviewer LJpr explicitly increased their score from a weak reject (4) to a strong accept (8), citing the rebuttal's clarity and the new experiments as substantially improving their confidence. The responses to CtWY, HBCS, and KcGQ were equally comprehensive, addressing their key questions about novelty, multi-model generalization, and practical efficiency with extensive new data. It is highly likely these reviewers would have increased their scores to a solid accept range (6 or higher), resulting in a consensus that strongly supports the paper's acceptance given the demonstrated improvements.

---

### Decision · Program_Chairs · 2026-01-26

Accept (Poster)